# Targets and genomic constraints of ectopic Dnmt3b expression

Yingying Zhang[1†], Jocelyn Charlton[1,2†], Rahul Karnik[1†], Isabel Beerman[1,3,4], Zachary D Smith[1], Hongcang Gu[5], Patrick Boyle[5], Xiaoli Mi[1], Kendell Clement[1], Ramona Pop[1], Andreas Gnirke[5], Derrick J Rossi[1,3,4], Alexander Meissner[1,2,5]*

[1]Department of Stem Cell and Regenerative Biology, Harvard University, Massachusetts, United States; [2]Department of Genome Regulation, Max Planck Institute for Molecular Genetics, Berlin, Germany; [3]Department of Pediatrics, Harvard Medical School, Massachusetts, United States; [4]Program in Cellular and Molecular Medicine, Division of Hematology/Oncology, Boston Children's Hospital, Massachusetts, United States; [5]Broad Institute of MIT and Harvard, Massachusetts, United States

*For correspondence: meissner@molgen.mpg.de

†These authors contributed equally to this work

Competing interests: The authors declare that no competing interests exist.

**Abstract** DNA methylation plays an essential role in mammalian genomes and expression of the responsible enzymes is tightly controlled. Deregulation of the de novo DNA methyltransferase DNMT3B is frequently observed across cancer types, yet little is known about its ectopic genomic targets. Here, we used an inducible transgenic mouse model to delineate rules for abnormal DNMT3B targeting, as well as the constraints of its activity across different cell types. Our results explain the preferential susceptibility of certain CpG islands to aberrant methylation and point to transcriptional state and the associated chromatin landscape as the strongest predictors. Although DNA methylation and H3K27me3 are usually non-overlapping at CpG islands, H3K27me3 can transiently co-occur with DNMT3B-induced DNA methylation. Our genome-wide data combined with ultra-deep locus-specific bisulfite sequencing suggest a distributive activity of ectopically expressed Dnmt3b that leads to discordant CpG island hypermethylation and provides new insights for interpreting the cancer methylome.

DOI: https://doi.org/10.7554/eLife.40757.001

## Introduction

DNA methylation is a major component of the epigenetic machinery that participates in the regulation of gene expression during development. The predominant targets for DNA methylation are cytosines in the CpG dinucleotide context (*Bird, 1986*). While the majority of CpGs in mammalian genomes are methylated, clusters of CpGs called CpG islands (CGIs) remain generally unmethylated, and are often located near or within housekeeping and developmental gene promoters (*Bird, 1986*; *Saxonov et al., 2006*). Pre-existing methylation patterns are propagated by the maintenance DNA methyltransferase DNMT1, and new modifications are generally added by the de novo methyltransferases DNMT3A and 3B (*Bestor, 2000*; *Charlton et al., 2018*; *Denis et al., 2011*; *Jeltsch, 2002*; *Xu and Corces, 2018*). Loss of these enzymes in mice causes prenatal (*Dnmt1* and *Dnmt3b*) or postnatal (*Dnmt3a*) lethality, which underscores their essential role in mammalian development (*Li et al., 1992*; *Okano et al., 1999*). DNMT3A and DNMT3B are structurally similar and appear to have redundant functions overall, although the knockout phenotypes are distinct and some specific targets are well established for DNMT3B, such as pericentromeric repeats and germline gene promoters (*Liao et al., 2015*; *Okano et al., 1999*). Both enzymes are highly expressed in early embryos, but only *DNMT3A* remains active in the adult and appears to be the major de novo methyltransferase involved in dynamic regulation of DNA methylation in somatic lineages (*Ziller et al., 2013*). In

contrast, levels of catalytically active *DNMT3B* decrease sharply during pluripotent stem cell differentiation as cells switch to an inactive isoform (*Gifford et al., 2013*; *Gordon et al., 2013*).

Deviations from the regulatory regime described above can lead to the aberrant expression of genes, genome instability, loss of imprinting and tumorigenesis (*Hamidi et al., 2015*; *Robertson, 2005*). In fact, deregulation of all three catalytically active human DNA methyltransferases is found across a wide range of diseases (*Hamidi et al., 2015*; *Robertson, 2005*) and mutations in both regulatory and catalytic domains are known contributing factors (*Jin et al., 2008*; *Klein et al., 2011*; *Winkelmann et al., 2012*; *Xu et al., 1999*; *Yan et al., 2011*). In contrast, it is not clear how aberrant expression of otherwise wild-type DNMTs, which is frequently observed in specific cancers, affects the genomic DNA methylation landscape (*Amara et al., 2010*; *Hayette et al., 2012*; *Jin et al., 2005*; *Kobayashi et al., 2011*; *Roll et al., 2008*). Although evidence exists that overexpression of DNMTs, especially DNMT3B, correlates with the epigenetic inactivation of tumor suppressor genes and tumor formation, primary tumors accrue substantial CGI methylation while the global average decays, and without temporal analysis, it cannot be ascertained whether global and local misregulation co-occur or if they represent distinct regulatory modes that arise independently (*Baylin and Jones, 2011*; *Ben Gacem et al., 2012*; *Butcher and Rodenhiser, 2007*; *Girault et al., 2003*; *Portela and Esteller, 2010*; *Roll et al., 2008*; *Steine et al., 2011*). Finally, even if DNMT3B overexpression is not a primary driver, the consequences of aberrant activity on cellular homeostasis during tumorigenesis remain incompletely understood and of direct relevance to human health.

From a mechanistic point of view, our understanding of the exact relationship between ectopic de novo methylation and other epigenetic modifications is limited, in particular for polycomb repressive complex 2 (PRC2) mediated H3K27me3, which is a repressive chromatin modification predominantly found at CGIs near developmental genes (*Lynch et al., 2012*; *Margueron and Reinberg, 2011*; *Tanay et al., 2007*). Previous work showed that DNA methylation and H3K27me3 are generally anti-correlated within CpG-rich regions but co-occur elsewhere in the genome (*Brinkman et al., 2012*; *Guo et al., 2014*; *Statham et al., 2012*). DNA methylation has also been suggested to directly interfere with PRC2 recruitment to CpG-rich sequences (*Jermann et al., 2014*). Conversely, loss of DNA methylation causes a global redistribution of H3K27me3 in both mouse embryonic stem cells (ESCs) and somatic cells (*Brinkman et al., 2012*; *Reddington et al., 2013*). This conditional antagonism between DNA methylation and H3K27me3 is quite unlike the constitutive antagonism between DNA methylation and H3K4me3, which is mediated by direct interaction of the ADD domain within DNMT3 and H3K4me3 (*Ooi et al., 2007*; *Otani et al., 2009*; *Zhang et al., 2010*). The interplay between DNA methylation and H3K27me3 has special relevance in cancer. Several studies have suggested that H3K27me3-enriched loci in ESCs are preferentially susceptible to gain of DNA methylation in many cancers (*Ohm et al., 2007*; *Schlesinger et al., 2007*; *Widschwendter et al., 2007*). CGIs that gained DNA methylation in a colon cancer cell line were depleted of H3K27me3 and switching from H3K27me3 to DNA hypermethylation was also observed at silenced gene promoters in human prostate cancer cells (*Brinkman et al., 2012*; *Gal-Yam et al., 2008*; *Statham et al., 2012*). However, coexistence of DNA methylation and H3K27me3 (dual modification) in promoter CGIs has also been reported in human cancer cell lines (*Gao et al., 2014*; *Statham et al., 2012*; *Takeshima et al., 2015*). Finally, we recently showed that the DNA methylation landscape in mouse extraembryonic tissues closely mirrors the altered landscape found across human cancer types, and that the establishment of this CGI hypermethylation is dependent on PRC2 and executed by DNMT3B (*Smith et al., 2017*). Because most of these observations have been made from cancer cell lines or primary tumor cells that are far removed from their initial transformation, elucidating the specific sequence of events underlying the gain of DNA methylation as it relates to H3K27me3 at CGIs remains of great interest.

To address these lingering questions, we systematically investigated the target spectrum of ectopic Dnmt3b expression using a tetracycline-inducible *Dnmt3b* transgenic mouse model to identify general rules and consequences of aberrant targeting. We specifically investigate why some CpG sites as well as entire CGIs are more prone to methylation when de novo methyltransferase activity is abnormally high, while others remain protected over long periods of time. Taken together, our system provides a unique opportunity to assess the effects of ectopic de novo methyltransferase activity on a genome-wide scale in a tissue-specific context and sheds new light on abnormal CGI hypermethylation as a result of *Dnmt3b* misregulation.

## Results

### Ectopic Dnmt3b expression leads to widespread CpG island hypermethylation

To better understand the relevance of de novo DNMT expression in cancer biology, we began by exploring the mechanism and frequency of DNMT3A and DNMT3B deregulation across different human cancer types. Specifically, we utilized a large, publicly available data set from The Cancer Genome Atlas (TCGA) (*Hoadley et al., 2014*) and analyzed the incidence of mutation and upregulation for each enzyme (*Figure 1A*). While Dnmt3b is generally not expressed in somatic tissues (*Figure 1—figure supplement 1A*), our analysis showed that nearly every tumor type contained samples where *DNMT3B* is upregulated (*Figure 1A*; *Figure 1—figure supplement 1B*) (*Duymich et al., 2016*). However, within each cancer type, only a fraction of tumors overexpress DNMT3B, while many more acquire CGI hypermethylation (*Figure 1B*). Nonetheless, the co-occurrence was high enough to justify a systematic investigation of DNMT3B deregulation in vivo. We began by generating mouse ESCs containing a heterozygous inducible *Dnmt3b1* construct and derived transgenic mice (*Beard et al., 2006*; *Linhart et al., 2007*) (*Figure 1—figure supplement 2A*). We then induced ectopic expression of Dnmt3b in the mice through doxycycline-supplemented drinking water and harvested six different tissues (liver, lung, muscle, intestine, kidney, heart) as well as select blood cell types (B cells, helper T cells, cytotoxic T cells, monocytes, granulocytes) from both induced and control (*Dnmt3b1* construct, non-induced) mice. To assess the level of DNA methylation with a focus on CGIs, we profiled these tissues using reduced representation bisulfite sequencing (RRBS) (*Meissner et al., 2005*; *Meissner et al., 2008*). Within a set of 7,467 CGIs covered in all samples, the median CGI methylation level consistently increases across all cell and tissue types, though both the percentage (6–15%) of hypermethylated CGIs and the average increase (2–9%) in mean CGI methylation vary upon Dnmt3b induction (*Figure 1C*). As RRBS also captures representative CpGs outside of CGIs, we determined the effects of Dnmt3b overexpression on various genomic features including promoters with high or low CpG density (HCPs and LCPs), CGI shores and gene bodies (exons, introns). We also included repetitive sequences; short interspersed nuclear elements (SINEs), long terminal repeats (LTRs) and long interspersed nuclear elements (LINEs) and found that all showed a similar trend of increased DNA methylation (*Figure 1—figure supplement 2B*). Finally, we specifically looked at regions enriched for H3K36me3, which are known to recruit DNMT3B through its PWWP domain (*Baubec et al., 2015*; *Dhayalan et al., 2010*; *Rondelet et al., 2016*), and found that these regions also gained DNA methylation despite beginning at already high levels (*Figure 1—figure supplement 2C*). In total, 152/4,260 and 222/9,824 H3K36me3-enriched regions showed significant additional gain in methylation (*P*-value <0.05, difference ≥0.2%, paired *t*-test) for kidney and liver respectively.

We next selected three tissues that showed a strong increase in CGI methylation (intestine, liver and kidney), confirmed the overexpression of Dnmt3b (*Figure 1—figure supplement 2D*), assessed the possible contribution of hydroxymethylation (*Figure 1—figure supplement 2E*), and determined their unique and shared hypermethylated CGIs. We did not observe any gross phenotypic changes in these tissues at the time of sample collection (*Figure 1—figure supplement 3A*), despite detecting a total of 4,839 differentially methylated CGIs across the three tissues (FDR *q*-value <0.05, methylation increase ≥0.2) from the 12,837 that were covered by RRBS (*Figure 1D*). The gain at CGIs was highly reproducible between both technical and biological replicates, suggesting a non-random increase at specific CGIs in each tissue (*Figure 1—figure supplement 3B*). We observed roughly equal numbers of differentially methylated regions (DMRs) at CGIs in intestine and liver, many of which were shared (*n* = 1,932). The overall number in kidney was lower, but a notable fraction was also shared with the other two tissues. Taken together, our system highlights a large, common set of targets as well as many unique, tissue-specific targets that may provide insight into the mechanism and consequences of aberrant DNMT3B-directed methylation.

### Ectopic Dnmt3b targets mostly silent and lowly expressed genes

Previous studies as well as our data show an extensive overlap of targets, with some additional tissue-specific effects, that led us to hypothesize that the level of gene expression may be one defining factor in the susceptibility of loci to ectopic Dnmt3 expression (*Athanasiadou et al., 2010*;

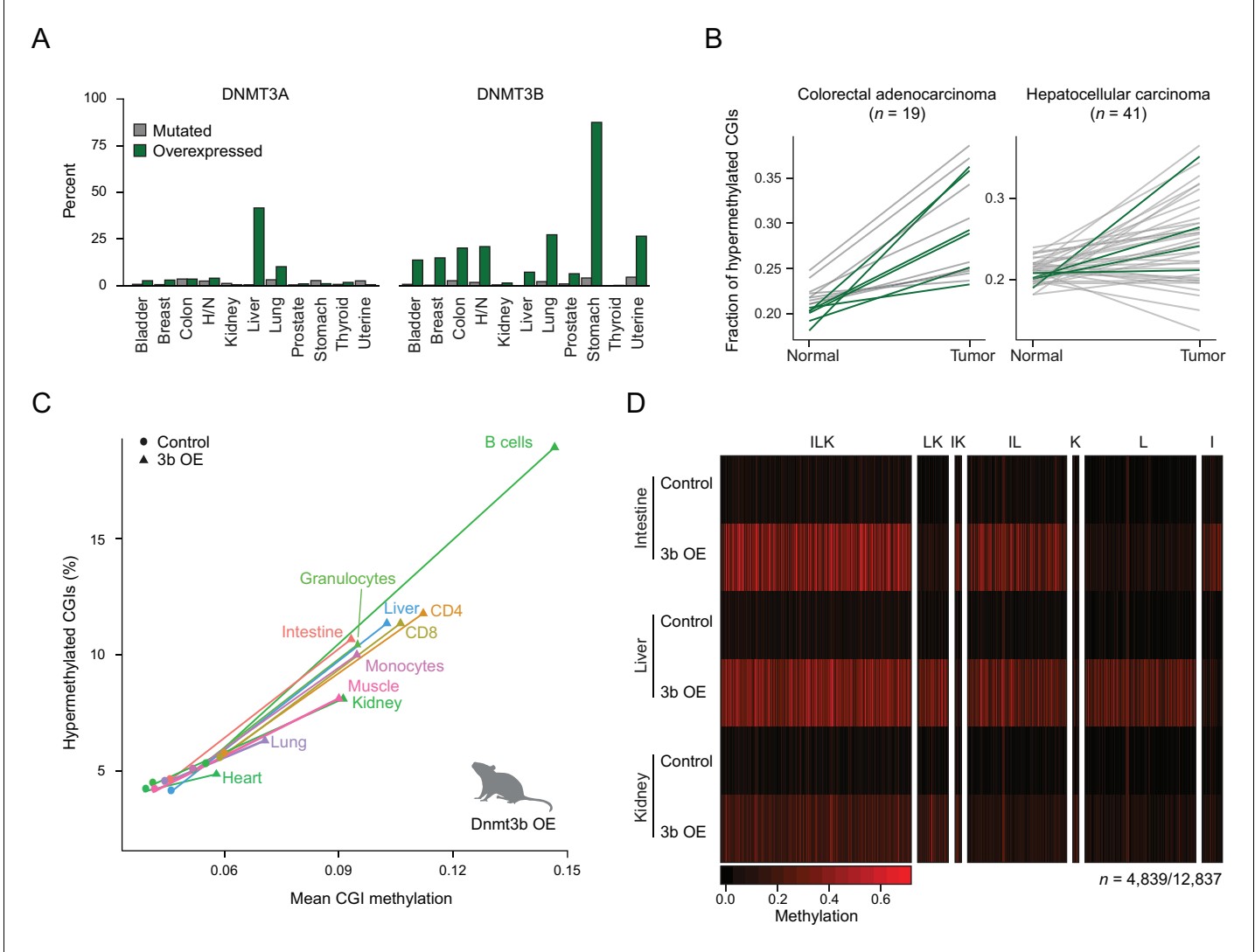

**Figure 1.** Dnmt3b overexpression in human cancer types and an ectopic mouse model. (A) Incidence of DNMT3A and DNMT3B mutation and overexpression in human cancers, based on data from The Cancer Genome Atlas (TCGA). H/N = head and neck. (B) Fraction of hypermethylated (defined as ≥0.3 increase in average CpG methylation) CpG islands (CGIs) in liver hepatocellular carcinomas and colorectal adenocarcinoma. Each line represents a pair of matched normal and tumor samples from a single patient. Patients that show overexpression of DNMT3A or DNMT3B (z-score ≥ 2) in the tumor sample are shown in dark green. (C) Mean CGI methylation vs percentage of hypermethylated CGIs (methylation ≥ 0.3) for 11 tissues harvested from age-matched control and Dnmt3b overexpression (3b OE) mice (0 – 1.5M dox). Data are based on RRBS profiles for matched CGIs (n = 7,467). (D) Heatmap of methylation levels at differentially methylated CGIs (FDR q-value <0.05, methylation difference of ≥ 0.2) in intestine, liver, and kidney upon 3b OE (3 – 6M dox). Each sample is the mean of two technical and two biological replicates. CGIs are clustered by their methylation status in the three tissues. Clusters are labeled with I (intestine), L (liver), or K (kidney) according to whether that cluster of DMRs was hypermethylated in the labeled tissue.

DOI: https://doi.org/10.7554/eLife.40757.002

The following figure supplements are available for figure 1:

**Figure supplement 1.** Normalized expression of mouse and human DNA methyltransferases across selected cell and tissue types as well as human DNMT isoform expression in TCGA samples.

DOI: https://doi.org/10.7554/eLife.40757.003

**Figure supplement 2.** Characteristics of ectopic Dnmt3b-induced CGI hypermethylation.

DOI: https://doi.org/10.7554/eLife.40757.004

**Figure supplement 3.** Tissues in which Dnmt3b is ectopically expressed are phenotypically normal and replicates show high reproducibility.

DOI: https://doi.org/10.7554/eLife.40757.005

*Gao et al., 2014*; *Meissner et al., 2008*). To test this more precisely, we isolated a number of cell types from the lymphoid lineage through fluorescence activated cell sorting (FACS) to minimize cellular heterogeneity within closely related cell types (*Bock et al., 2012*). We selected hematopoietic stem cells (HSCs), multipotent progenitors (MPPs, Flk2 positive), common lymphoid progenitor (CLP), differentiated helper T cells (CD4+) and cytotoxic T cells (CD8+) to look for close coupling between aberrant methylation and dynamic expression changes (*Figure 2a*, *Figure 2—figure supplement 1A*). It is worth noting that ectopic Dnmt3b expression did not affect the overall representation of stem, progenitor and differentiated cell types (*Figure 2—figure supplement 1B*). Similar

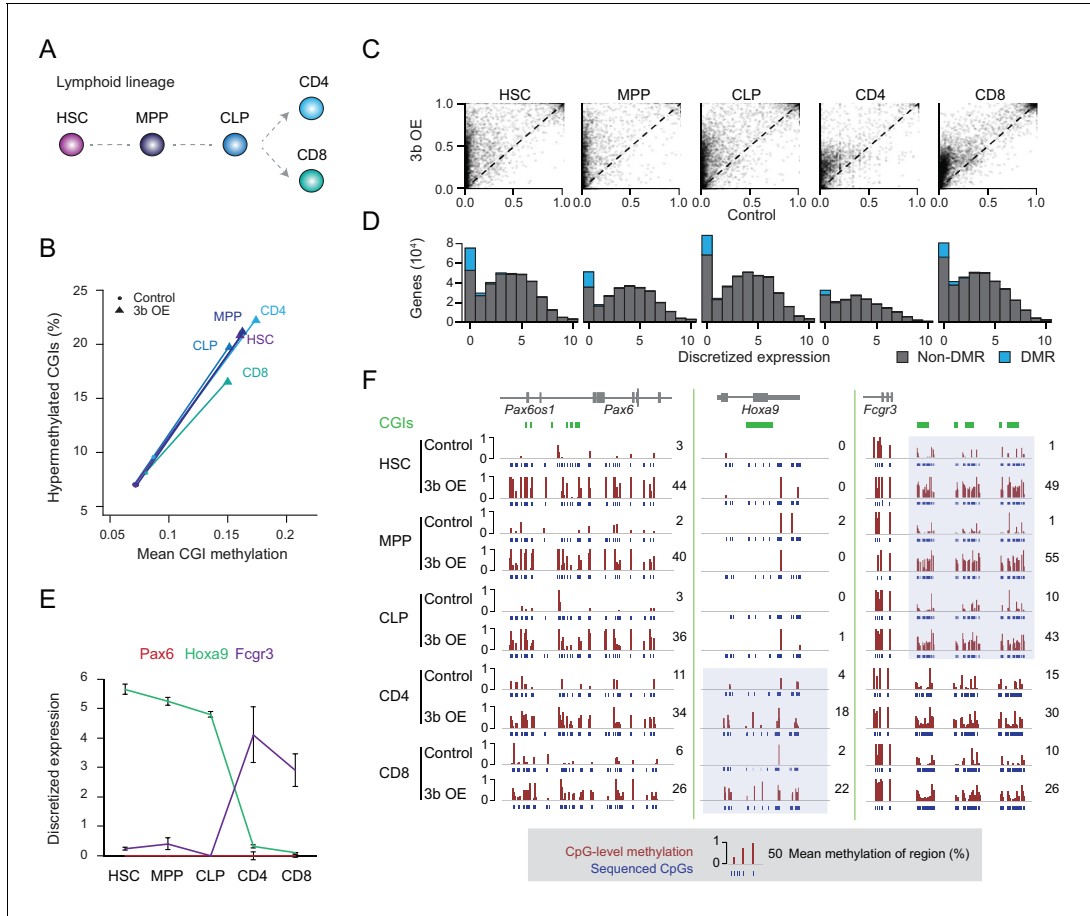

**Figure 2.** Ectopic DNMT3B targets predominantly silent and lowly expressed genes. (**A**) Schematic of selected hematopoietic cell types isolated from Dnmt3b induced and control mice (0 – 3M dox). HSC: hematopoietic stem cell, MPP: multipotent progenitor, Flk2 positive, CLP: common lymphoid progenitor, CD4/8+: T lymphocytes. (**B**) Mean CGI methylation vs percentage of hypermethylated CGIs (methylation ≥0.3) for the different blood cell types sorted from age-matched control and Dnmt3b overexpression (3b OE) mice (0 – 3M dox). Data are based on RRBS profiles for matched CGIs (*n* = 9,283) (**C**) Scatter plots of DNA methylation level for consistently covered CGIs (*n* = 14,584) in induced and control mice. (**D**) Number of differentially methylated promoters in each cell type stratified by normalized expression value of the associated gene. Data are taken from normal lymphoid differentiation (*Bock et al., 2012*). (**E**) WT expression levels of *Pax6*, *Hoxa9* and *Fcgr3* across all five hematopoietic cell types assayed. Data are taken from normal lymphoid differentiation (*Bock et al., 2012*). (**F**) Matching DNA methylation data as genome browser tracks and mean methylation (number on the right) for the displayed regions around *Pax6* (chr2:105,506,372–105,523,591), *Hoxa9* (chr2:105,506,372–105,523,591) and *Fcgr3* (chr1:172,986,522–173,013,977). The relevant DMRs at the *Hoxa9* and *Fcgr3* promoters in the blood cells are highlighted. For the CGIs located near to *Fcgr3*, comparing control and induced CLP samples generated highly significant *P*-values ($2.5 \times 10^{-11}$ to $4.03 \times 10^{-22}$). Although the level of methylation gained at CGIs in CD4+ cells was clearly lower than in CLP cells (mean difference of 0.19 vs 0.37), the *P*-values for CD4+ 3bOE vs control were still significant ($6.0 \times 10^{-4}$ to $7.83 \times 10^{-6}$). Blue boxes highlight the difference in methylation gain that correlates with change in gene expression.

DOI: https://doi.org/10.7554/eLife.40757.006

The following figure supplement is available for figure 2:

**Figure supplement 1.** Analysis of blood cell population frequencies and prevalence of DMRs by promoter class.

DOI: https://doi.org/10.7554/eLife.40757.007

to the solid tissues, we found hypermethylated CGIs in response to *Dnmt3b* expression in every cell type analyzed (*Figure 2B,C*). More CGIs showed hypermethylation (*q*-value <0.05, methylation gain >0.3) in HSCs and progenitor cells (HSC: *n* = 1,611, MPP: *n* = 1,117, CLP: *n* = 1,380) than in fully differentiated T cells (CD4+: *n* = 492, CD8+: *n* = 567), although this may be associated with the relatively strict threshold as CD4+ and CD8+ cells appear to show less increase and already slightly higher baseline levels at targeted CGIs (*Figure 2C*). It is also possible that some of the aberrant CGI methylation may either be lost during cell differentiation or lineage progression may selectively expand subsets of HSCs or early progenitors with lower methylation levels at these targets. We next binned genes according to their expression level in wild-type blood cells (*Bock et al., 2012*) and counted the number of differentially methylated promoters in each expression class. DMRs were consistently enriched for the most lowly expressed genes, an overall trend that was maintained when gene promoters were divided into HCPs and LCPs (*Figure 2D*, *Figure 2—figure supplement 1C*).

To more closely inspect the relationship between DNA methylation targeting and minimal transcription in the context of our extended lineage data, where the number of methylated CGIs decreases from stem/progenitor to terminally differentiated cell, we selected three genes (*Pax6*, *Hoxa9* and *Fcgr3*) with distinct expression patterns. PAX6, a key transcription factor in eye and nervous system development, is not expressed in any of the blood cell types (*Osumi et al., 2008*; *Shaham et al., 2012*). In contrast, Hoxa9 is expressed in all the stem/progenitor cells, but not in the differentiated T cells. Fcgr3 shows the opposite pattern and is induced in the differentiated T cells (*Figure 2E*). In line with our prior observation, we found a strong relationship between the expression state (in WT cells) and DNA methylation levels after ectopic Dnmt3b induction. While the *Pax6* locus remained unmethylated (median methylation 2 – 11%) in control mice, we found increased DNA methylation (median methylation 26 – 44%) across all cell types upon Dnmt3b induction (*Figure 2F*). Alternatively, the *Hoxa9* promoter gained methylation in helper T cells and cytotoxic T cells, but was not methylated in HSCs or progenitor cells, where it functions to enhance HSC activity and suppress lymphoid differentiation (*Thorsteinsdottir et al., 2002*). Hoxa9 is only lowly expressed in T cells, where it was abnormally methylated after Dnmt3b induction. Lastly, at the *Fcgr3* promoter, we found the highest level of methylation in the stem/progenitor cells, with CD4+ and CD8+ cells still showing some methylation in both control and 3b OE cells, but with a much smaller differential (*Figure 2F*). We do find other rare examples of hypermethylated CGIs losing methylation during differentiation and an even smaller fraction gaining expression (*Figure 2 – Figure Supplement 1D*), but cannot distinguish whether the decrease in methylation is related to passive proliferation-dependent loss, active removal of DNA methylation through upstream factors, or a selective shift in the population (note the median levels in HSCs are high but not 100%, indicating possible heterogeneity).

Nonetheless, we can conclude that de novo methylation through ectopic Dnmt3b occurs predominantly at the promoters of already silent/lowly expressed genes, which may explain both the shared and unique targets that we observed above. However, since not all silent genes gain DNA methylation, additional factors may influence the target spectrum.

## H3K4me3 shields CGIs from aberrant DNA methylation

To generalize the observation that targeting of ectopic DNMT3B occurs within a subset of silent genes, we examined the gene expression status of hypermethylated promoters in the livers of induced mice (*Figure 3A*). After confirming this relationship, we hypothesized that additional modifications to target chromatin may explain why some silent loci remain protected from de novo methylation, while others do not. To explore this, we correlated the ectopic Dnmt3b RRBS data with H3K4me3 and H3K27me3 chromatin immunoprecipitation-sequencing (ChIP-seq) data of liver tissue from control mice and found that DMRs displayed significantly lower levels of H3K4me3 relative to non-DMRs at HCP and LCPs. In contrast, HCPs that gained methylation had significantly higher H3K27me3 levels than non-DMR HCPs (*Figure 3B*). When we annotated all genomic CGIs by their chromatin state as H3K4me3, H3K27me3, both (bivalent) (*Bernstein et al., 2006*) or neither, and calculated the percentage of CGIs in each class that gained DNA methylation, we found that H3K27me3–only CGIs were much more likely to be hypermethylated than those that were bivalent or enriched for H3K4me3 (*Figure 3C*).

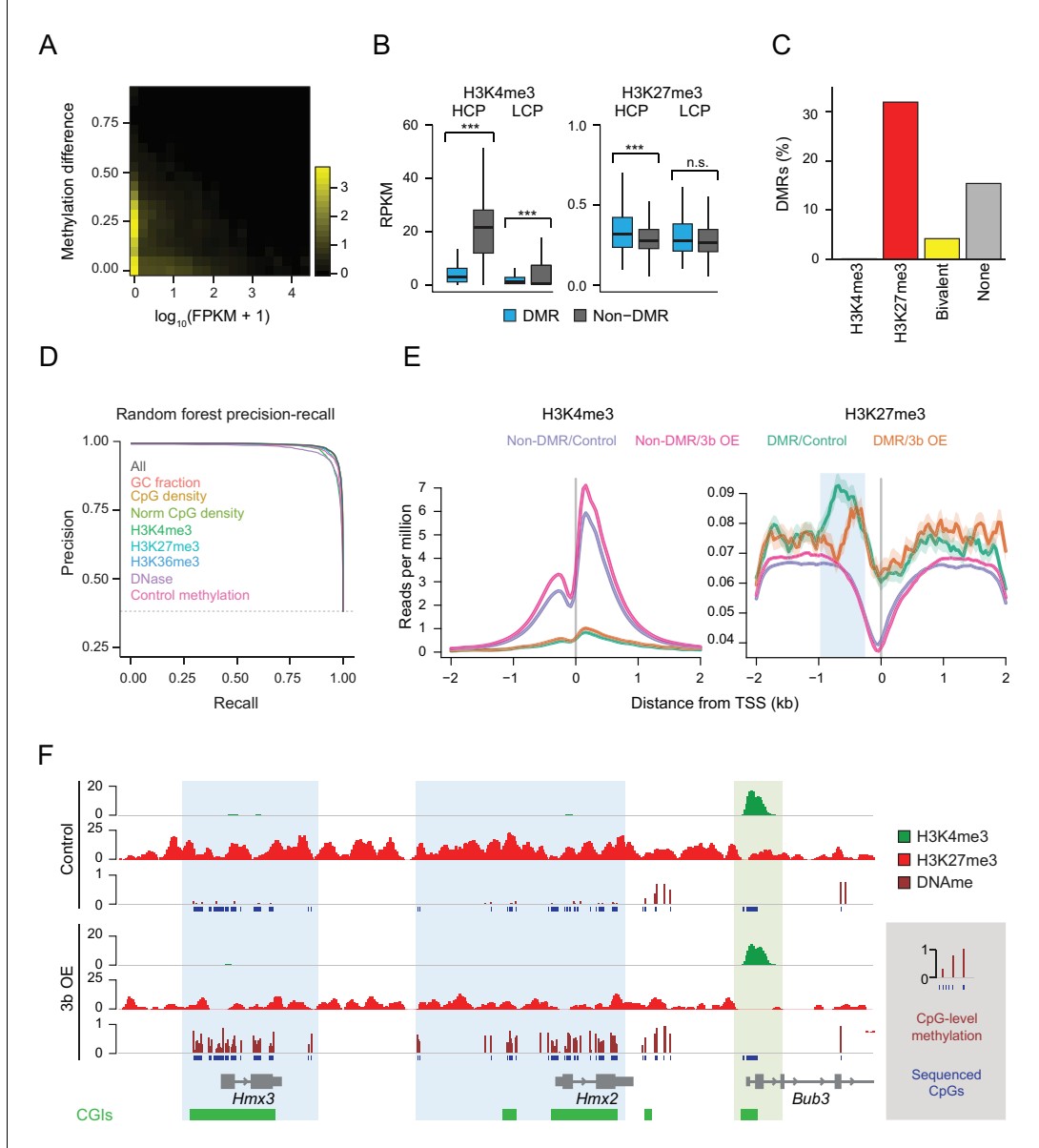

**Figure 3.** Underlying chromatin landscape defines the ectopic DNMT3B target spectrum.  (A) Two-dimensional density plot of methylation gain at promoter CGIs vs. expression of corresponding genes in mouse liver (3–6M dox). (B) H3K4me3 and H3K27me3 enrichment at DMR vs non-DMR promoters prior to *Dnmt3b* induction (mouse liver). Promoters are classified into high CpG density promoters (HCP) and low CpG density promoters (LCP). The distributions of the chromatin marks were compared using a two-sample *t*-test. H3K4me3 was significantly lower at DMRs both for HCPs (*P*-value = $5\times10^{-258}$) and LCPs (*P*-value = $3\times10^{-46}$). H3K27me3 was slightly, but significantly (*P*-value = $9\times10^{-8}$) enriched at HCP DMRs. Boxes display the interquartile range and whiskers extend to the most extreme data point that is no more than 1.5 times the interquartile range; the bold line indicates the median value. (C) Fraction of CGIs that are differentially methylated when Dnmt3b is overexpressed in liver, stratified into those enriched (FDR *q*-value < 0.05) for H3K4me3, H3K27me3, H3K4me3 and H3K27me3 (bivalent) or neither mark. (D) Prediction recall curves generated using a random forest classifier predicting the likelihood of a given DMR being a DMR for liver tissue (3 – 6M). 'All' uses all listed features as classifiers. For each labeled feature, the respective curve shows the shift in prediction when that feature is removed from the classifier. The top three most influential factors affecting prediction were H3K4me3, DNase and control methylation level. (E) Mean read density plots of H3K4me3 and H3K27me3 at DMR HCPs and non-DMR HCPs centered on transcription start sites (TSS) in liver from control and Dnmt3b induced mice (0 – 3M dox). The blue shading highlights the decrease in H3K27me3 upstream of the TSS under Dnmt3b induction. (F) Genome browser tracks for H3K27me3, H3K4me3 and DNA methylation (RRBS) over a representative genomic region (chr7:138,683,072–138,708,076). CGI DMRs overlapping the *Hmx3* and *Hmx2* loci are highlighted in light blue. A nearby CGI within the *Bub3* promoter region appears protected and shows H3K4me3 enrichment in both control and induced (0 – 3M dox) livers (green highlight).

DOI: https://doi.org/10.7554/eLife.40757.008

*Figure 3 continued on next page*

*Figure 3 continued*

The following figure supplement is available for figure 3:

**Figure supplement 1.** Cross-validation performance of logistic regression at predicting hypermethylated CGIs.
DOI: https://doi.org/10.7554/eLife.40757.009

To quantify the predictive value of H3K4me3 and H3K27me3 with respect to hypermethylation at CGIs, we built a random forest classifier using H3K4me3 and H3K27me3 enrichment, as well as six other features that may influence prediction: CpG density, normalized CpG density, GC fraction, H3K36me3 enrichment, DNase signal and control methylation level. Prediction was then assessed using five-fold cross-validation. Using all features, we were able to recover 96.7% DMRs at a false discovery rate of 5%, demonstrating the high predictive impact of all features in combination (*Figure 3D*). The performance of the random forest classifier was very similar to, but generally better than, five other classification methods we applied using the same set of features (*Figure 3—figure supplement 1A*). We next removed each parameter from the model in turn to assess its contribution to DMR detection, and noted that H3K4me3, DNase and control methylation level displayed the greatest decrease in performance (*Figure 3D*, *Figure 3—figure supplement 1B*). As H3K4me3 and DNase signal are correlated with each other and anti-correlated with control methylation levels and H3K27me3 (*Figure 3—figure supplement 1C*), this analysis shows that CGIs with very low levels of methylation and enriched for H3K4me3 in open chromatin are the least susceptible to methylation gain.

We next examined the distribution of H3K4me3 and H3K27me3 at transcription start sites (TSSs) for DMRs and non-DMRs and found DMRs were substantially enriched for H3K27me3 and depleted of H3K4me3 (*Figure 3E*). Moreover, in the ectopic Dnmt3b livers, we found no net effect on H3K4me3 at DMR promoters, but a depletion of H3K27me3 (*Figure 3E*). For example, CGIs overlapping *Hmx2* and *Hmx3* both gained DNA methylation, and H3K27me3 decreased around these loci (*Figure 3F*). In contrast, the promoter CGI of *Bbu3* was completely protected and showed enrichment for H3K4me3. These dynamics suggest that regions with H3K4me3 enrichment remain protected from gain of DNA methylation despite high levels of ectopic DNMT3B, in agreement with previous studies (*Zhang et al., 2010*). In contrast, regions that have H3K27me3 are normally unmethylated, but are generally more susceptible to aberrant methylation.

## Hypermethylation is rapidly induced at H3K27me3-enriched CGIs in MEFs

Previous results have demonstrated that DNA methylation and H3K27me3 are mostly anti-correlated at CGIs, at least within the steady state of continuously renewing cell lines (*Brinkman et al., 2012*). For instance, in HCT116 cells, regions that have gained DNA methylation compared to normal colon tissue show a clear depletion of H3K27me3 (*Brinkman et al., 2012*; *Statham et al., 2012*). We therefore wanted to utilize our system to identify the series of events that led to the regulatory switch from facultative silencing through H3K27me3 to perhaps more stable repression by DNA methylation. To do so, we performed a set of experiments in mouse embryonic fibroblasts (MEFs) isolated from E13.5 transgenic embryos for improved experimental control and higher temporal resolution. We induced *Dnmt3b* expression in MEFs for one and seven days (1 d and 7 d) and profiled them using RRBS (*Figure 4A*). The overall number of hypermethylated CGIs was comparable to our in vivo samples ($n$ = 908 and 3,160 for 1 and 7 d, respectively, FDR $q$-value <0.05 and gain of methylation $\geq$0.2). Within one day of induction, a large number of CGIs gained methylation, and both the number and methylation level continued to increase after longer exposure (*Figure 4B*). Our in vivo observations indicate that the target spectrum of DNMT3B appears to be largely dictated by gene expression and the underlying presence or absence of H3K4me3 and H3K27me3. To explore this further, we classified genes into highly and lowly expressed, and then examined the incidence of DMRs at their promoters depending on their chromatin status, specifically the presence of H3K4me3, H3K27me3, both/bivalent or neither (*Figure 4c*, *Figure 4—figure supplement 1A*). Most highly expressed HCP-directed genes (2,378/2,485) were enriched for H3K4me3 and were generally protected from de novo methylation: no genes from this group gained methylation after 1 day, and only 1% showed any change in methylation after 7 days of induction. Lowly expressed HCP-directed

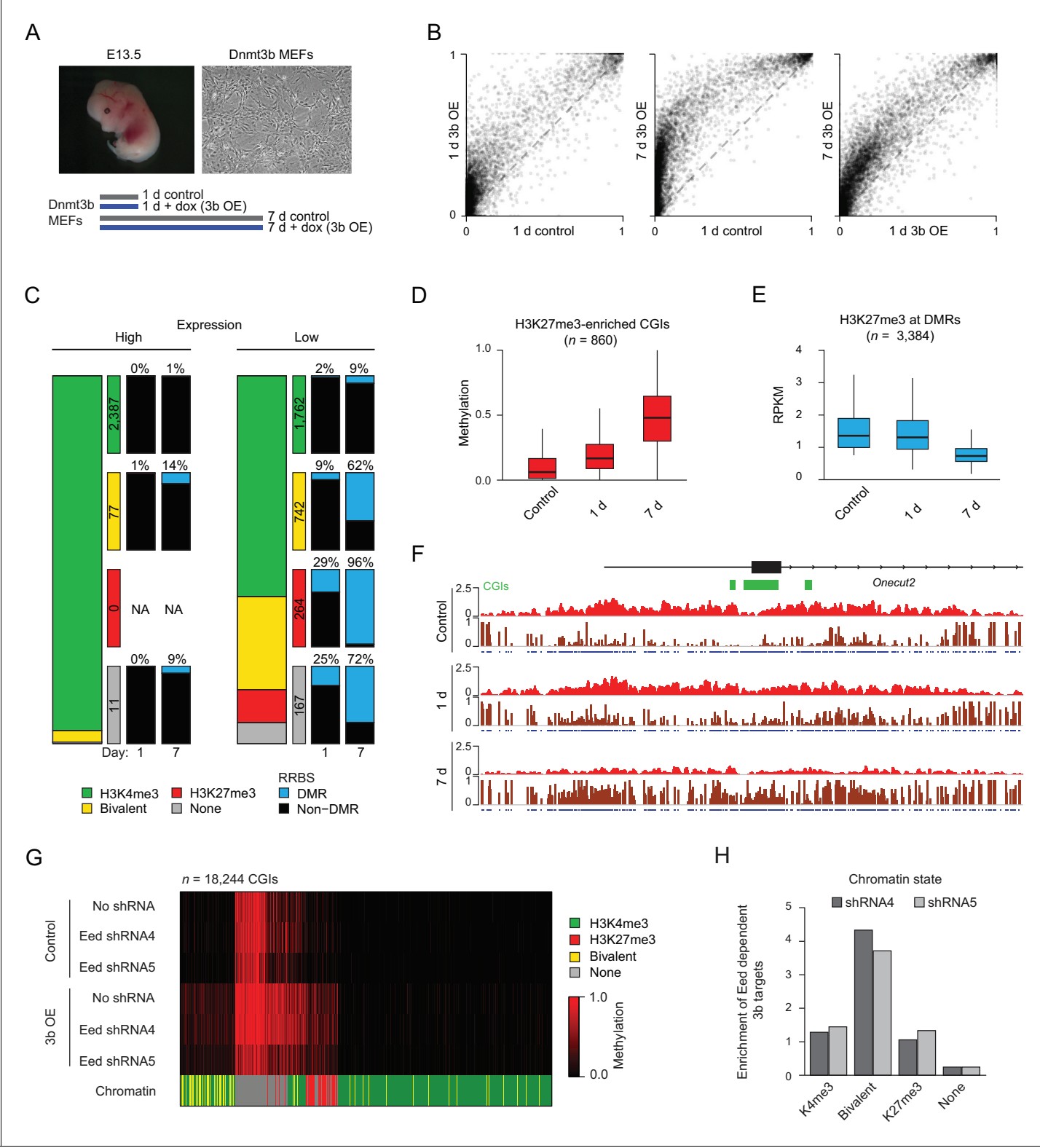

**Figure 4.** *Dnmt3b* induced hypermethylation co-occurs with and eventually displaces H3K27me3. (**A**) Inducible mouse embryonic fibroblasts (MEFs) were isolated from E13.5 embryos. To minimize the effect of cell culture and study early dynamics, we used a short (one day: 1 d) and extended (seven day: 7 d) induction and collected time-matched uninduced controls. (**B**) Scatter plot showing the pairwise comparison of CGI methylation (*n* = 14,513) in control and induced MEFs (1 d and 7 d). (**C**) Correlation of differential methylation after 1 d and 7 d dox with gene expression, H3K4me3 and H3K27me3 status for HCPs in MEFs. Stacked bar plots show the division of CGIs by their chromatin state with color-coded boxes labeling the number

*Figure 4 continued on next page*

*Figure 4 continued*

of CGIs in each category. The boxes to the right show a representation of the percentage of CGIs methylated after 1d and 7d induction. (D) Boxplots showing the distribution of DNA methylation levels for all CGIs that are enriched for H3K27me3 (based on the ChIP-BS-seq data; not necessarily DMRs), from MEFs overexpressing Dnmt3b for 1 d and 7 d. Boxes display the interquartile range and whiskers extend to the most extreme data point that is no more than 1.5 times the interquartile range; the bold line indicates the median value. (E) Boxplots showing H3K27me3 enrichment at significant CGI-DMRs in MEFs after inducing Dnmt3b for 1 d and 7 d using ChIP-BS-seq data. Boxes display the interquartile range and whiskers extend to the most extreme data point that is no more than 1.5 times the interquartile range; the bold line indicates the median value. (F) IGV genome browser tracks for H3K27me3 and ChIP-BS-seq data at a representative genomic region (chr18: 64,488,248–64,511,818) in control and induced (1 d and 7 d) MEFs. (G) Control and Dnmt3b overexpression (3b OE) MEF samples were profiled by RRBS in the presence of no shRNA and two different shRNAs targeting the PRC2 component Eed. The heatmap displays mean methylation over all covered CGIs. For each CGI, the chromatin status for H3K4me3 and H3K27me3 is displayed. The Eed shRNA was introduced into WT MEFs before 3bOE was induced via lentiviral transduction. (H) For dox-treated samples, the genomic location of 3b OE targets that are not methylated when a given Eed shRNA is also present compared to the background distribution of CGIs in each category.

DOI: https://doi.org/10.7554/eLife.40757.010

The following figure supplements are available for figure 4:

**Figure supplement 1.** Ectopic Dnmt3b expression in MEFs.

DOI: https://doi.org/10.7554/eLife.40757.011

**Figure supplement 2.** *Eed* knockdown in MEFs.

DOI: https://doi.org/10.7554/eLife.40757.012

genes also remained protected from methylation as long as they were enriched for H3K4me3. Alternatively, silent bivalent and K27me3-only HCPs were methylated early with 9% and 29% of loci on day 1 and 62% and 96% at day 7, respectively. For genes with neither H3K4me3 nor H3K27me3, 9% of highly expressed genes and 72% lowly expressed genes gained promoter methylation by day 7. These results strongly support our previous in vivo conclusions that ectopic *Dnmt3b* is only able to target the promoters of lowly expressed genes, especially those enriched for H3K27me3. Similar results were also observed for genes with LCPs (*Figure 4—figure supplement 1A*).

## DNMT3B-induced methylation transiently co-occurs with H3K27me3

Prior results indicate that DNA methylation and H3K27me3 are typically non-overlapping at CGIs, and diminished H3K27me3 with a simultaneous gain of intermediate levels of DNA methylation could indicate either transient co-occurrence of both modifications at the same nucleosome or heterogeneity within the population. To investigate the sequence of events that leads to gain of DNA methylation and loss of H3K27me3, we performed H3K27me3 ChIP-bisulfite-sequencing (ChIP-BS-seq) on MEFs prior to and following ectopic Dnmt3b expression (*Brinkman et al., 2012*; *Statham et al., 2012*). Although H3K27me3 enrichment appeared globally reduced in the control MEFs after seven days (*Figure 4—figure supplement 1B*), dox-induced samples demonstrated further depletion of H3K27me3 specifically at DMR-CGIs concurrent with gain of de novo methylation, while control (not CGI, methylated) regions remained enriched (*Figure 4—figure supplement 1C*). H3K27me3-containing chromatin showed widespread gain of methylation at day 1 that increased to a higher level on day 7 (*Figure 4D–F*). As such, DNMT3B can engage targets with H3K27me3 and methylate the surrounding DNA, which may eventually interfere with H3K27me3 maintenance and trigger its subsequent loss.

Since ectopic *Dnmt3b* expression mostly targets already silent loci, we had expected minimal effects on the transcriptional program as long as cells are not changing their cellular state. To test this, we performed RNA-seq on MEFs after induction (7 d) and in matched control cells (*Figure 4—figure supplement 1D*). As expected, transcriptional changes were minimal. Altogether, we saw only 39 genes that were significantly downregulated and 13 genes that were upregulated upon Dnmt3b overexpression (adjusted $P$-value <0.05, fold change >2), not including Dnmt3b itself. Interestingly, the most highly upregulated gene was *Rab6b*, an oncogene, which was upregulated almost 4-fold upon *Dnmt3b* overexpression. Most downregulated genes gained methylation at their promoters; however, no clear correlation between DNA methylation level changes at the promoter and gene expression was observed for upregulated genes (*Figure 4—figure supplement 1E*).

## Knockdown of Eed has a limited impact on MEF CGI hypermethylation

Given that H3K27me3 does not appear to obstruct ectopic Dnmt3b activity at target loci, we wanted to investigate the alternative possibility of H3K27me3 or PRC2 being involved in ectopic DNMT3B recruitment. We used two independent shRNAs to knockdown Eed, an essential core component of PRC2, in MEFs before inducing ectopic Dnmt3b expression for seven days (*Figure 4—figure supplement 2A–C*) and collecting samples for RRBS analysis. It is important to note that although we were able to reduce H3K27me3 levels by ~10 fold, it was not entirely depleted, meaning that some CGIs within the population of cells could still be modified with H3K27me3. Nonetheless, despite global H3K27me3 depletion, both Eed KD MEF lines still showed widespread CGI hypermethylation that increased to similar mean levels as our prior control MEFs (*Figure 4G*, *Figure 4—figure supplement 2D*), suggesting that PRC2 activity may not be essential for ectopic targeting in this context. Upon closer inspection, we found that while one Eed KD (shRNA-4) showed very similar CGI targeting and mean methylation to WT MEFs (no KD: *n* = 5,943, mean = 0.22, shRNA-4: *n* = 5,883, mean = 0.21), the other (shRNA-5) had a slightly reduced impact (*n* = 4,510, mean = 0.16). CGIs that were no longer targeted after Eed KD in dox-treated MEFs (*n* = 647 and *n* = 1,645) showed a ~4 fold enrichment for bivalent chromatin (*Figure 4H*), raising the possibility that the remaining presence of H3K4me3 may continue to shield CGIs from de novo methylation after the loss of H3K27me3. Together, these findings suggest that H3K27me3 and/or PRC2 are likely not required for inducing CGI hypermethylation in this ectopic Dnmt3b overexpression model, although the complex or its modification appears essential for the targeting of selected CGIs in other contexts (*Smith et al., 2017*).

## Dnmt3b expression induces local methylation discordance

In the above experiments, we often saw changes in DNA methylation at CGIs on the order of 10–40%. These low to intermediate levels of methylation could be a reflection of cellular heterogeneity or a truly intermediate level of methylation that is present in every cell. In order to distinguish between these two scenarios, we investigated the emergence of DNA methylation at a representative CGI locus that contains 50 CpG sites near the TSS of the *Uncx* gene. We amplified the region from bisulfite converted DNA isolated from induced MEFs, generated amplicon libraries and sequenced these libraries deeply, obtaining between 450,000 and 650,000 reads for each library (*Figure 5A*). Consistent with previous results, we found a clear increase in DNA methylation at all CpG sites from day 1 to day 7, with median methylation at this locus increasing from 0.02 in the control cells to 0.08 after 1 day of treatment, and to 0.18 after 7 days of treatment. The DNA methylation pattern of the region for 10,000 representative molecules, each likely representing a unique epiallelic measurement, indicate substantial intermediate methylation within the majority of reads (*Figure 5B*). The absence of highly methylated fragments argues against discrete cellular heterogeneity. Rather, methylation is observed at multiple CpGs within the locus in each cell, as can be seen from the distribution of per-read (implying per-cell) methylation (*Figure 5C*). Similar DNA methylation patterns are also observed in tissues overexpressing Dnmt3b (*Figure 5—figure supplement 1A*).

   To further quantify this phenomenon, we counted the number of epialleles, defined as a mixture of methylation patterns of a group of adjacent CpG sites with variable frequencies in a cell population (*Landan et al., 2012*). The total number of epialleles increased from 3,207 in the control cells to 14,469 (1 d) and to 16,915 (7 d). It is interesting that the number of epialleles increases substantially after the first day of treatment, but then seems to level off as methylation continues to increase. We would expect that the number of epialleles decreases as the number of methylated CpGs increases beyond 25 (half of the total in the amplicon), but we see a decrease in the number of epialleles even when the number of methylated CpGs is less than 15 (*Figure 5C*). This decrease in the number of epialleles suggests that, while the initial CpGs methylated are chosen somewhat randomly, subsequent methylation is more frequently placed on CpGs based on the existing methylation pattern. Alternatively, some CpGs may be more stably propagated once methylated, which would effectively limit the diversity of ensuing epialleles as additional CpGs within the locus are modified. Indeed, certain CpG sites within the locus appear more susceptible while others seemed resistant to methylation gain, with the percentage of reads methylated at each specific CpG varying from 2% to 54% across the sequence. Five consecutive CpG sites (#22 to #26) showed extremely low methylation level compared to other CpG sites in MEFs (7 d) (*Figure 5A*), as well as in intestine and liver

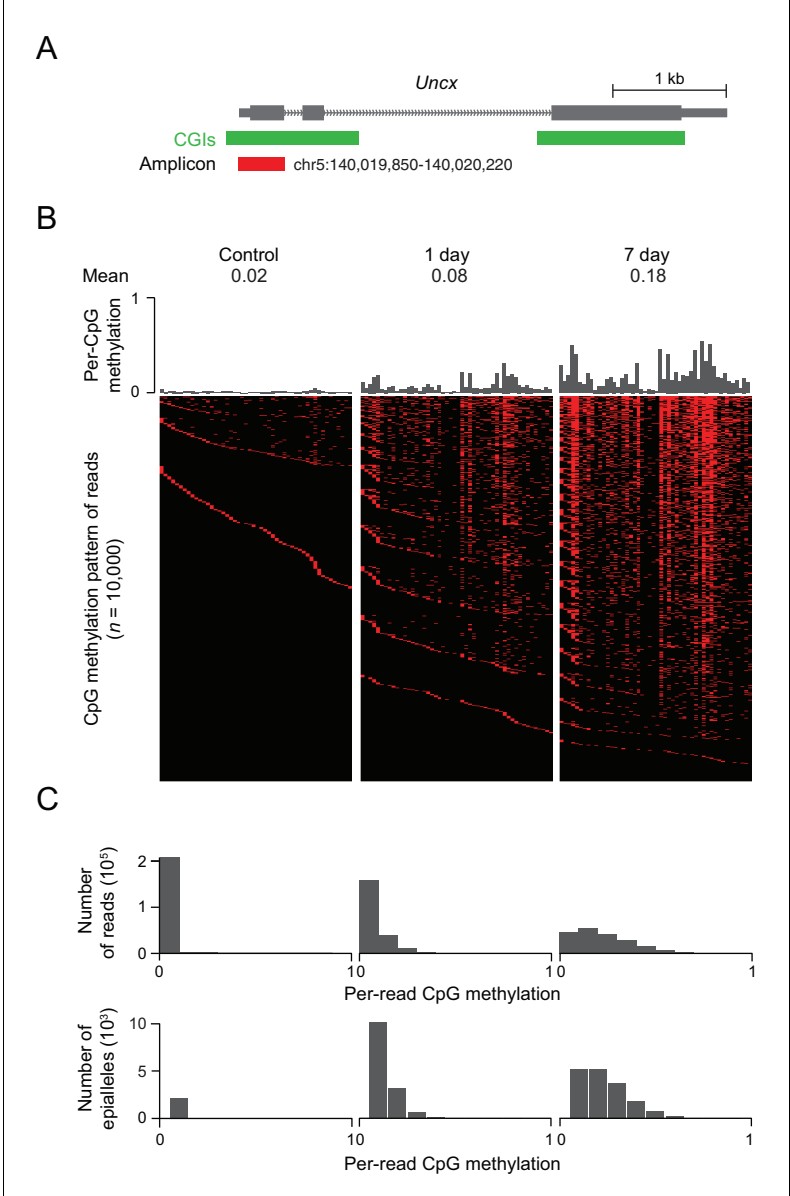

**Figure 5.** DNMT3B deposits heterogeneous methylation at CGIs. (**A**) We selected the CGI promoter of the *Uncx* gene for ultra-deep bisulfite sequencing. The target amplicon covers 50 CpGs for which the mean methylation levels in control and induced (1 d and 7 d) MEFs are shown. (**B**) Top: Per CpG methylation over the amplicon region (not drawn to genomic scale). Bottom: Heatmap of individual reads (*n* = 10,000 randomly selected) showing the patterns (epialleles) of methylated and unmethylated CpGs within individual molecules. Black indicates unmethylated and red methylated C's in the CpG dinucleotide context. (**C**) Distribution of per-read methylation and distribution of the number of epialleles across the number of methylated CpGs per molecule for all reads (*n* = 93,636–147,954).

DOI: https://doi.org/10.7554/eLife.40757.013

The following figure supplement is available for figure 5:

**Figure supplement 1.** Ultra-deep amplicon-based bisulfite sequencing in mouse tissues and motif analysis.
DOI: https://doi.org/10.7554/eLife.40757.014

(*Figure 5—figure supplement 1A*). It appears, based on underlying motif analysis, that these CpGs may remain hypomethylated due to the presence of transcription factor binding sites (*Figure 5—figure supplement 1B and C*). Finally, the preferential gain at certain sites may also be associated with the reported flanking sequence preference for DNMT enzymes (*Emperle et al., 2018*; *Handa and*

*Jeltsch, 2005*). In both our control and Dnmt3b overexpressing cells, we noticed reads where only two or a few CpGs were methylated and that these CpGs were rarely in proximity to one another. We therefore looked at the pairwise correlation of CpGs in phase within our amplicon sequencing reads as a function of the genomic distance between them (*Figure 5—figure supplement 1D*) and observed low correlation levels (maximum 0.37), with neighboring CpGs showing very little coordinated gain in methylation (*Figure 5—figure supplement 1D*). This indicates that DNMT3B may act focally at each CpG, either by distributive association/dissociation with the DNA, or by a processive (continuous) walk with skipping of CpGs along its path.

## Ectopic Dnmt3b targeting shares only some features with cancer CGI hypermethylation

In order to connect our mechanistic findings to the TCGA results in *Figure 1*, we explored CGI methylation levels in three cancer types from tissues that match those collected in our study and observed similar trends (*Figure 6A*). Closer inspection of the mouse data revealed a clear net increase in methylation over the center of liver DMR CGIs, but also a high differential between control and induced samples at the CGI periphery (±2 kb, *Figure 6B*). CGI shores have previously been shown to be hypermethylated in cancers (*Irizarry et al., 2009*) and we found several orthologous human tumor suppressor genes or biomarkers that gained methylation at promoter-proximal CGIs and shores when Dnmt3b was overexpressed in mouse tissues (*Figure 6—figure supplement 1A*). Our genome-wide results also confirm a selected subset of targets that was previously reported in the context of ectopic Dnmt3b expression in *APC^{min}* mice, which lead to an increase in adenoma formation (*Linhart et al., 2007*) (*Figure 6—figure supplement 1B*). However, we have not performed a long-term induction in wild-type mice to explore whether the hypermethylation alone would be sufficient to trigger precursor lesions or tumor formation.

We next directly compared methylation levels at orthologous CGIs and noted a general trend of similar CGIs gaining methylation in our ectopic Dnmt3b tissues, chronic lymphocytic leukemia (CLL) and early postimplantation mouse extraembryonic ectoderm (ExE) (*Figure 6—figure supplement 1C*). These were enriched for CGIs that display bivalent chromatin in human ESCs, which are considered a subset of highly targeted ubiquitous CGIs across human cancers (*Smith et al., 2017*) (*Figure 6—figure supplement 1D*). Moreover, read level methylation showed an increase in local discordance between nearby CpGs on the same molecule (*Figure 6—figure supplement 2A*), which has been previously associated with poor outcome in CLL (*Landau et al., 2014*). However, upon closer inspection of mouse control and Dnmt3b overexpressing B cells to human normal B cells and CLL, we see that the majority of bivalent (81%) and H3K27me3-only (93%) CGIs are targeted by ectopic Dnmt3b, while only 12% and 21% are targeted in CLL, rendering the overlap high, but significance low (*Figure 6C*). This extreme difference may be linked to the fact that our comparison included the most highly methylated induced tissue (B cells, *Figure 1C*) and a relatively hypomethylated cancer, and may not be as pronounced for a different tissue. Overall, we conclude that even though the ectopic system methylates a large proportion of cancer targets (84% orthologous CLL CGIs in total), this may in part be due to the high expression levels of Dnmt3b rather than a shared mechanism. This is also reflected in the higher levels of differential methylation per CGI in Dnmt3b overexpressing B cells compared to CLL (*Figure 6D*).

Furthermore, while our system showed an overall increase in methylation at 100 bp tiles regardless of the original methylation level (but highest for tiles with low to intermediate original methylation), the distribution of methylation changes in colorectal cancer (TCGA) showed similar increases at lowly methylated genomic regions, but a loss of methylation at intermediate and highly methylated genomic regions, in keeping with the general global hypomethylation reported in cancer (*Figure 6E*). This also held true for CLL (RRBS) and ExE, which also both display global hypomethylation (*Figure 6—figure supplement 2B*). Together, these features suggest that our ectopic system shares some common targets and possible mechanisms with human cancer, including preferential targeting of CGIs that have H3K27me3 or bivalent chromatin. However, it does not fully recapitulate the global methylation trends that characterize the cancer methylome.

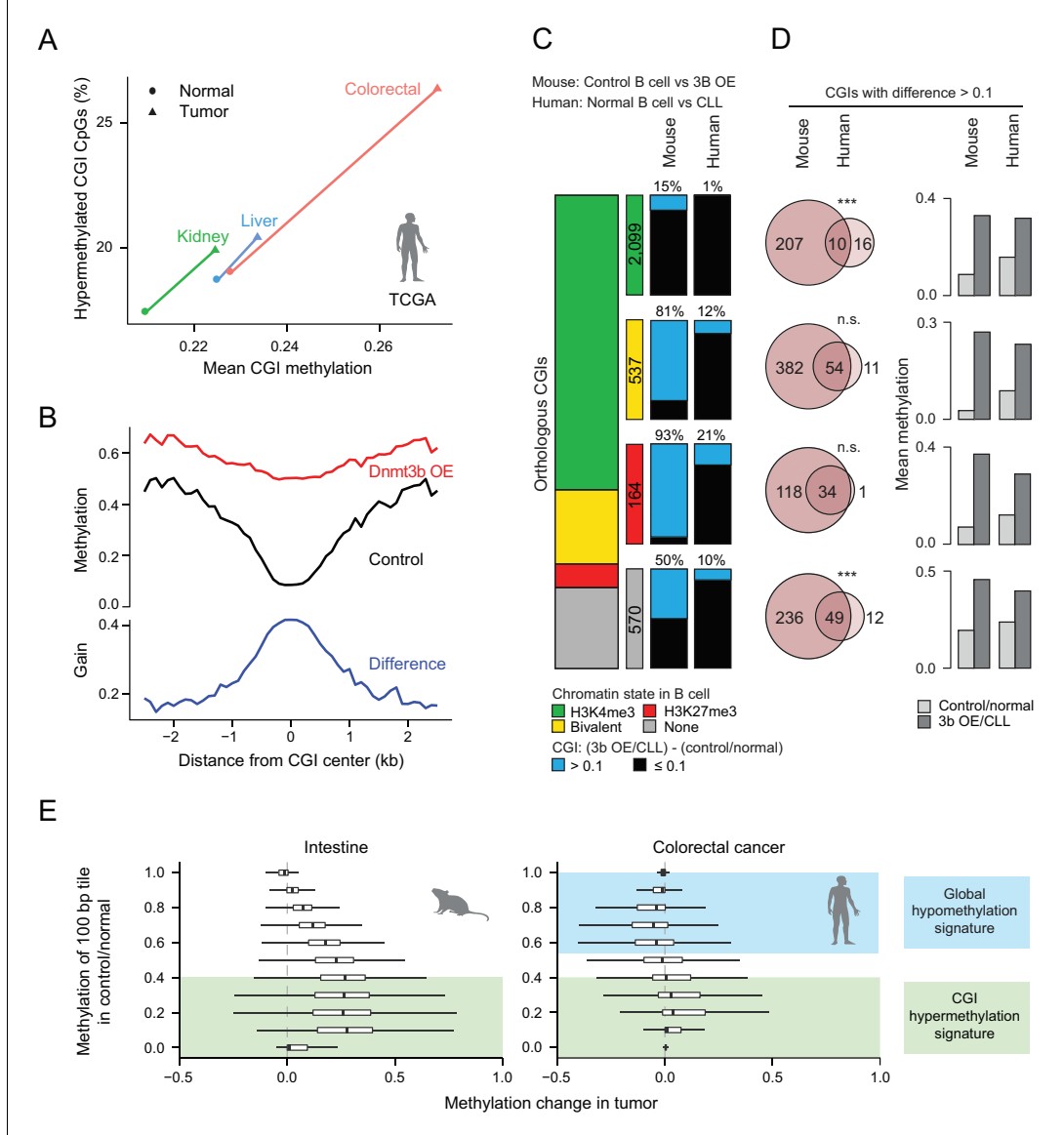

**Figure 6.** Comparison of features between the ectopic Dnmt3b expression and human cancers. (A) Mean CGI hypermethylation in matched human samples (cancer and normal). Tumors show an increase in the mean methylation levels of CGIs (x-axis) driven by an overall increase in the fraction of hypermethylated CpGs (beta value >0) in CGIs (y-axis). Data are based on Illumina Infinium HumanMethylation450 BeadChip profiles generated by TCGA. (B) Composite plot of CGI DMRs in 3b OE (3–6M dox) and control mouse liver demonstrates that targeted methylation within the center of CGIs exhibits a higher net increase from control values than is observed in the periphery (CGI shores). (C) For orthologous CGIs that showed matched chromatin modification in both human and mouse normal B cells (n = 3,370), the proportion that fall into the categories K3K4me3, bivalent, H3K27me3 and neither are displayed as a stacked barplot with the number in each category displayed in the color-matched box to the right. Within each category, the number of CGIs that gain methylation (>0.1) in mouse 3b OE B cells (0 – 1.5M dox) vs control (left) or human CLL vs normal B cells (right) is displayed. (D) For CGIs that gained methylation (>0.1), the number of unique and overlapping CGIs as well as the significance of the overlap based on their hypergeometric distribution is displayed (left). The mean level of methylation for these same CGIs in control/normal and 3b OE/CLL is shown (right). (E) Comparison of global DNA methylation changes that occur in human colon cancer and in mouse intestine when *Dnmt3b* is overexpressed. The mean methylation level of 100 bp tiles was calculated for CpGs in control/normal samples (y-axis) and compared to the change in methylation in the dox-treated/cancer sample (x-axis). In colorectal cancer, DNA methylation decreases in hypermethylated regions, while some lowly methylated regions gain methylation. In the mouse 3b OE model, lowly methylated and intermediately methylated regions gain methylation, but the global cancer-specific hypomethylation is not observed.

DOI: https://doi.org/10.7554/eLife.40757.015

The following figure supplements are available for figure 6:

**Figure supplement 1.** Ectopic Dnmt3b expression shows similarities to cancer CGI hypermethylation.

*Figure 6 continued on next page*

*Figure 6 continued*

DOI: https://doi.org/10.7554/eLife.40757.016

**Figure supplement 2.** Increased methylation discordance but lack of global hypomethylation in ectopic Dnmt3b tissues.

DOI: https://doi.org/10.7554/eLife.40757.017

## Discussion

In mammals, somatic cell types generally maintain a highly methylated genome with the exception of CGIs that remain constitutively unmethylated regardless of identity and expression state (*Smith and Meissner, 2013*). Even during the massive wave of de novo methylation in the early postimplantation epiblast, where Dnmt3a and Dnmt3b are very highly expressed, the unmethylated state of CGIs is preserved (*Smith et al., 2017*). Following global remethylation of the genome, DNMT3B activity is subdued and DNMT3A appears generally responsible for the more local dynamics at distal *cis*-regulatory elements in later developmental transitions. In contrast, a notable fraction of human cancers show deregulated DNMT3B expression. With little known about the consequence of elevated abnormal activity on the genome or cell physiology, we utilized an inducible mouse model to derive a set of general rules that help predict the possible targets and downstream effects.

Specifically, we found that DNMT3B preferentially targets CGIs near silent gene promoters, which tend to be enriched for H3K27me3, a repressive histone modification catalyzed by PRC2. Our results also show that de novo methylation by ectopically expressed Dnmt3b can transiently co-exist with H3K27me3-modified histones, although H3K27me3 appears to be lost as DNA methylation continues to increase, which raises some interesting and mechanistically relevant questions. Although a CpG-density-dependent effect has previously been suggested (*Brinkman et al., 2012*), our *Uncx* CGI amplicon demonstrates a modest but population-wide increase that is targeted to distinct CpGs in different cells, making it unclear how such a minor increase in epigenetic signal could obstruct PRC2. Alternatively, the increased frequency of ectopic DNMT3B enzyme contacts required for its distributive activity may interfere with PRC2 or somehow limit its H3K27me3 maintenance function. In this scenario, all DNMT3B isoforms, including catalytically inactive DNMT3B3, may be sufficient to cause depletion of H3K27me3. There is some evidence that non-catalytic DNMT3B isoforms bind DNA and influence the location or activity of binding partners. Specifically, DNMT3B3 has been shown to act as an accessory protein to stimulate gene body methylation (*Duymich et al., 2016*), aid de novo methylation of target loci in cancer cells and is depleted upon 5-aza treatment (*Weisenberger et al., 2004*). The lack of H3K27me3 maintenance upon increase in DNA methylation might also be mediated by PRC2 associated proteins. While the core subunits of PRC2 (Eed, Suz12 and Ezh1/2) do not bind DNA (*Weisenberger et al., 2004*), interaction with DNA is thought to be mediated through associated proteins such as AEBP2, JARID2 or the Polycomb-like proteins (PCLs) (*Li et al., 2017*). Recently, MTF2 was shown to be essential for PRC2 recruitment in mouse ESCs, with tertiary DNA structure demarcating target versus non-target CGIs (*Perino et al., 2018*). As MTF2 was further shown to be methylation-sensitive, it is possible that DNMT3B overexpression may interfere with its binding and eventually disrupt H3K27me3 maintenance. Finally, the lack of DNA methylation at H3K27me3 modified CGIs has also been linked to the presence of FBXL10 (also called KDM2B, NDY1, JHM1B, CXXC2) (*Boulard et al., 2015*). In ESCs, nearly all promoter-CGIs are bound by FBXL10 (*Boulard et al., 2015*), and deletion in mouse ESCs results in hypermethylation of ~25% usually unmethylated promoter-CGIs that are co-bound with PRC1 and PRC2. Increased methylation of normally unmethylated CGIs was also observed in mouse epiblast upon *Fbxl10*-disruption, further demonstrating its critical role in vivo (*Smith et al., 2017*; *Boulard et al., 2015*). Taken together, these results add to our understanding of the complex and multi-layered machinery that preserves the canonical, unmethylated status of CGIs across most somatic cell types.

Despite the ability of Dnmt3b to target CGIs modified with H3K27me3, we found that not every H3K27me3-only CGI was hypermethylated, which is also true for placental progenitors as well as human cancers. While this may be partially due to slightly increased expression of certain CGI promoters, CGI hypermethylation susceptibility likely involves a more intricate regulatory system. To elucidate these mechanisms, we feel much can still be learned from further studying in vitro systems including ectopic Dnmt3b expression and relating them to developmental or disease states where CGIs appear to be methylated as part of a concerted regulatory transition. At present, we do not

know whether the observed susceptibility of CGIs to hypermethylation is associated with high ectopic expression in our system or whether Dnmt3b could readily target CGIs even at low expression levels.

Our results suggest that the general increased activity of DNMT3B alone is sufficient for de novo methylation across a characteristic and predictable set of CGIs. However, CGI hypermethylation rarely increases beyond intermediate levels (for instance in liver, 90% of CGIs show less than 20% mean methylation), suggesting a high degree of heterogeneity, which is also found across cancer types (*Landau et al., 2014*; *Smith et al., 2017*). In conclusion, we provide several new insights into the underlying mechanisms of aberrant CGI methylation that is a hallmark of most cancers.

## Materials and methods

### Generation of *Dnmt3b* ESCs and transgenic mice

KH2 ESCs were cultured in standard serum/LIF conditions as described previously (*Meissner et al., 2009*). Targeting of the *Dnmt3b* cDNAs into the *Col1A1* locus of KH2 cells was conducted using the gene targeting kit from *Open Biosystems* based on the original publication (*Beard et al., 2006*). Briefly *Dnmt3b1* cDNAs were cloned into the pBS31'-RBGpA vector and electroporated (2 pulses at 500 V and 25 µF on Bio-Rad Gene Pulser II) into the KH2 cells along with pCAGGS FLPe recombinase plasmid. Clones with successful integration of the targeting vector were identified though hygromycin (140 µg/ml) selection for 10 days.

To assess induction in the targeted ESCs, 1 mM retinoic acid (Sigma) was added for 6 days to induce differentiation and as a consequence decrease the endogenous *Dnmt3b* expression level.

The *Dnmt3b* targeted ESCs were used to generate mice through diploid blastocyst injections (HSCI Genome Modification Facility). Chimeras were backcrossed to C57BL/6 females (The Jackson Laboratory) to obtain F1 heterozygous transgenic mice. For genotyping, DNA was isolated from tail biopsies and subjected to PCR using primers and conditions as previously described (*Beard et al., 2006*).

For transgene induction, mice were fed 1 mg/mL doxycycline in the drinking water supplemented with 10 mg/mL sucrose for the specified duration.

### MEF culture and shRNA constructs

MEFs were isolated from Embryonic day (E)13.5 embryos of control and heterozygous *Dnmt3b* transgenic mice and cultured in knockout Dulbecco's modified Eagle's medium (KD DMEM, Life Technologies) supplemented with 10% fetal bovine serum (Seradigm), as previously described (*Meissner et al., 2009*). Early passage (P3) MEFs were expanded, induced (2 µg/ml) for one or seven days and then collected for analysis (RRBS, amplicon and ChIP-seq).

Two shRNAs, shRNA-4 (GTATGTTTGGGATTTAGAA) and shRNA-5 (GCAACAGAGTAACCTTATA) targeting different exons of *Eed* were cloned into a pSicoR-ef1a-GFP vector for lentivirus production. For the combined knockdown and *Dnmt3b* overexpression experiment, the *Dnmt3b* cDNA was linked to 2A-mCitrine and cloned into the FUW-tetO vector.

### Virus production and infection

Virus was generated by co-transfecting lentiviral constructs, pCMV-dR8.2 viral packaging plasmid, and the pCMV-VSVG viral coat plasmid (Addgene) into HEK293T cells using FuGENE6 reagent (Promega). Supernatants containing virus particles were collected at 48 hr and 72 hr post transfection, and were passed through a 0.45 µm filter. For infection, MEFs were incubated with the viral solutions containing 8 µg/ml Polybrene for 5–7 hr at 37°C. After virus infection for 48 hr, cells were split and dox (2 µg/ml) was added to the cell culture medium. On day 7 post dox induction, the cells with fluorescence were isolated on a BD Biosciences FACSAria II cell sorter. Genomic DNA from cells was extracted for RRBS.

### Tissue harvesting and blood cell sorting

Transgenic *Dnmt3b* heterozygous mice were induced as described above and samples collected after different induction times/windows (see text for details). Tissues (liver, intestine, kidney, muscle,

lung etc.) were harvested from at least two independently induced mice (and matching controls) and snap frozen in liquid nitrogen.

The blood stem, progenitor and differentiated cells were purified from control and *Dnmt3b* mice (0 – 3M dox, three biological replicates) as previously described (*Bock et al., 2012*). Briefly, HSC, MPP1, MPP2, CLP, CMP, GMP and MEP were harvested from the bone marrow of adult mice and stained with combinations of FITC-CD34, PE-Flk2, APC-eFlour780-CD117, APC-Ly-6A/E, Biotin-line-age (Ter119, CD4, CD3, CD8, Mac1, GR1 and IL7ra) with PacificOrange-streptavidin secondary, Percp-Cy5.5-CD16/CD32 and PacificBlue-Il7ra. T lymphocytes (CD4+, CD8+), B-cells, granulocytes and monocytes were harvested from peripheral blood of adult mice and stained using a staining cocktail of PerCpCy5.5-Ter119, FITC-Gr1, Pe-Cy7-Mac1, PacificBlue-CD8, APC-Cy7-B220, APC-CD71, Biotin-CD4 and PacificOrange-streptavidin. Cells with the surface markers used in a previous study (*Bock et al., 2012*) were sorted on a BD Biosciences FACSAria II cell sorter, and reanalysis showed >98% purity of the isolated cell populations.

## Genomic DNA extraction and RRBS library preparation

Flash-frozen mouse tissues and cell pellets were lysed in 300 – 600 µl DNA lysis buffer (10 mM Tris-HCl pH 8.0, 10 mM EDTA, 10 mM NaCl and 0.5% wt/vol SDS) supplemented with 50 ng/µl DNase-free RNase (Roche) and 1 µg/µl proteinase K (NEB) at 55°C overnight. After phenol:chloroform extraction and ethanol precipitation, DNA was re-suspended in elution buffer (10 mM Tris-HCl pH 8.0, 1 mM EDTA). 100 ng genomic DNA was used to make RRBS libraries using the methods as previously described (*Boyle et al., 2012*).

## Hydroxymethylation analysis

To assess hydroxymethylation at selected CpGs, we used the Quest 5-hmC Detection Kit (Zymo Research) according to manufacturer's instructions.

## Gene expression analysis

For Western blotting analysis, proteins extracted from cultured cells were separated by electrophoresis using NuPAGE 4–12% Bis-Tris gel and transferred to a PVDF membrane using the iBlot Dry Blotting System (Invitrogen). Histone proteins for western blotting were extracted from cell pellets using the protocol as previously described (*Shechter et al., 2007*). The following antibodies were used for western blotting analysis: anti-DNMT3B (monoclonal, Abcam, ab52A1018), anti-beta ACTIN (polyclonal, Abcam, ab8227), anti-histone H3 (polyclonal, Abcam 1791), anti-H3K27me3 (polyclonal, Millipore 07–449).

For real time RT-qPCR, RNA was isolated from cultured cells or tissues using RNeasy Mini Kit (Qiagen) and treated with DNase. 1 µg RNA was converted to cDNA using the First-Strand cDNA Synthesis Kit (Qiagen). qPCR was performed in triplicates using SYBR Green master mix (Applied biosystems) with gene specific primers listed in the *Supplementary file 1*. Relative fold changes were determined by the $2-\Delta\Delta CT$ method. The housekeeping genes *Hprt* and *Pgk1* (QuantiTect primers from Qiagen) were used as internal control for normalization.

For RNA-seq, 100 ng RNA extracted from each tissue and cell type was used to make the libraries using TruSeq RNA Sample Preparation Kit (Illumina). Libraries were sequenced on *Bestor (2000)* instruments using standard protocol. For expression in sorted blood cells, we used publically available microarray data from *Bock et al. (2012)*.

## ChIP-seq libraries

Liver harvested from *Dnmt3b* mice (0 – 3M dox) were first chopped into fine pieces, then crosslinked with 1% formaldehyde for 15 min at room temperature, quenched with 125 mM glycine for 5 min, and washed with PBS containing protease inhibitor (PI, Roche). The pellets were suspended in cell lysis buffer (1% SDS, 10 mM EDTA and 50 mM Tris-HCl, pH 8.1, PI) and homogenized using the TissueLyser tissue disruption system (Qiagen). The nuclei were further extracted using nuclear lysis buffer (0.1% SDS, 10 mM Tris–HCl, pH 7.5, 1% NP-40, 0.5% sodium deoxycholate, PI), and sonicated on Branson Sonifier (model S-450D) to achieve chromatin fragments ranging 200 and 700 bp. Chromatin was incubated with the H3K27me3 (Millipore 07 – 449) and H3K4me3 antibody (Millipore, 07 – 473) respectively at 4°C overnight. The clean-up step and the library construction method were

as previously described (*Ziller et al., 2015*). The sequencing of libraries was performed on the Illumina HiSeq 2000.

For MEFs, 10 million cells were crosslinked with 1% formaldehyde for 10 min at room temperature, quenched with 125 mM glycine for 5 min, and washed with PBS containing PI. The pellets were first suspended in cell lysis buffer (1% SDS, 10 mM EDTA and 50 mM Tris-HCl, pH 8.1, PI), then in nuclei lysis buffer (0.1% SDS, 10 mM Tris–HCl, pH 7.5, 1% NP-40, 0.5% sodium deoxycholate, PI) to obtain the chromatin. The remaining procedure is the same as described for liver ChIP-seq.

### ChIP-BS-seq library construction

ChIP-BS-seq libraries were generated as previously described (*Brinkman et al., 2012*) with a few modifications. Briefly, 40 – 80 ng of ChIP DNA was used for preparing ChIP-BS libraries. After end repair, DNA fragments were phenol:chloroform:isoamyl alcohol (25:24:1) extracted and ethanol precipitated. The DNA pellets were dissolved in EB buffer, and then subjected to terminal 3' adenylation in a 30 µl reaction containing 3 µl of 10x T4 DNA ligase buffer, 2.5 µl of Klenow fragment (Klenow Fragment (3'agmeexo-, NEB) and 1.5 µl of 10 mM dATP. The A-tailing reaction was incubated at 37 ˚C for 30 min followed by 20 min at 65 ˚C. Adapter ligation was conducted directly in the A-tailing reaction by adding Illumina Truseq indexed adapter (1:20 dilution) and T4 ligase (2 million units/ml, NEB). After overnight ligation at 16 ˚C, the ligation reaction was heated at 65 ˚C for 20 min to inactivate T4 ligase. Adapter-equipped DNA fragments were extracted using 1.5X Agencourt AMPure XP beads and subsequently eluted into 20 µl EB buffer.

Bisulfite conversion of the adapter equipped DNA was performed as described previously (*Brinkman et al., 2012*) and the second round of bisulfite-converted DNA fragments were eluted to 40 µl of EB buffer. To minimize duplicated reads in sequencing results, we performed an analytical PCR experiment to optimize the PCR cycle number with conditions as following: 95 ˚C for 2.5 min, varied numbers of PCR cycles (10, 13, 16, 19) - 95 ˚C for 30 s, 60 ˚C for 30 s, and 72 ˚C for 1 min, followed by 7 min final extension at 72 ˚C. The final DNA libraries were enriched in 8 × 25 µl of PCR reactions each composing of 2.5 µl of 10x PCR buffer, 0.5 µl PfuTurbo Cx Hotstart DNA Polymerase, 1.5 µl of 2.5 µM of Truseq PCR primers, 0.25 µl of 100 mM dNTP, and 3 µl of bisulfite converted DNA. PCR products were purified using a Qiagen MinElute PCR purification kit and the library DNA was eluted to 20 µl of EB buffer.

To remove PCR primer-dimmers and normalize library DNA size ranges across different samples, the eluted DNA was run on a 2.5% NuSieve (3:1) agarose gel and DNA fragments with a size range of 250 – 700 bp was cut and subsequently purified using the Qiagen MinElute Gel purification kit. The final ChIP-BS libraries were quantified using an Agilent Bioanalyzer with a High Sensitivity DNA Chip and pooled equally based on molar concentrations of each library. The sequencing of libraries was performed on the Illumina HiSeq 2000.

### Deep sequencing of amplicons from bisulfite converted genomic DNA

100 ng genomic DNA from MEFs or tissues was treated with bisulfite using EpiTect Fast Bisulfite Conversion Kits (Qiagen). PCR was performed using primers specific for CGIs in the promoter region of Uncx gene with the bisulfite converted DNA as template (forward primer 5'-TGATGTTGATAAAG TAA AGY(C/T)G-3', reverse primer 5'-CTCCAACCTACCTACAAACTTAAA-3'). The PCR products (408 bps) were further purified and ligated to Illumina Truseq indexed adapter (1:30 dilution). The libraries generated from each PCR product were mixed at equal molar concentration and the sequencing was performed on a MiSeq instrument using MiSeq Reagent Kits v2 (Illumina, 2 × 250 cycles).

### Definitions of genomic features

CGIs were defined as genomic regions with GC content greater than 0.5, length greater than 200 bp, and ratio of observed CpGs to expected CpGs greater than 0.6. CGI shores were defined as 2,000 bp upstream and downstream of CGIs, but not overlapping neighboring CGIs.

We used Ensembl transcription start sites applied to the mm9 build (Ensembl release 67) of the mouse genome (*Cunningham et al., 2015*), and defined promoters as 500 bp and 500 bp downstream of the TSS. Promoters overlapping CpG islands were classified as high CpG density promoters (HCPs), while the rest were classified as low CpG density promoters (LCP).

Repeat annotations were downloaded from the UCSC genome browser site (*Kent et al., 2002*) and are based on RepeatMasker (Smit AFA).

## TCGA data processing and analysis

All TCGA datasets were generated by the TCGA Research Network (http://cancergenome.nih.gov/).

TCGA mutation and expression data was downloaded using the 'cgdsr' R library (http://www.cbioportal.org/cgds_r.jsp) to access the cBioPortal for Cancer Genomics (*Cerami et al., 2012*; *Gao et al., 2014*). All non-synonymous changes to the protein sequences were counted as 'Mutated'. Cases with RNA-seq *z*-score values greater than or equal to three were designated as 'Upregulated'.

TCGA methylation data from Illumina 450K arrays was downloaded from the Broad GDAC (Broad Institute TCGA Genome Data Analysis Center (2015): Firehose 0.4.4, Broad Institute of MIT and Harvard, doi:10.7908/C15 × 282W).

## RRBS data processing and analysis

Raw sequencing reads were aligned to the mm9 build of the mouse genome using MAQ (*Li et al., 2008*). As in previous studies we used our custom Python scripts (https://github.com/meissnerlab/zhang_elife_2018; copy archived at https://github.com/elifesciences-publications/zhang_elife_2018) to call methylation at each CpG, computed as the ratio of methylated reads to total reads at that genomic position.

Methylation for a feature was calculated as the median of the methylation of individual CpGs within the feature, weighted by coverage. At least 2 CpGs covered at 5X were required for each feature-level methylation calculation.

Differentially methylated regions were detected by using a weighted t-test (*Bland and Kerry, 1998*) with CpGs covered at 5x within the region used as the set of samples for each condition. The *t*-test was weighted by the coverage at each CpG. False discovery rate/q-values were computed using the R qvalue package (*Storey and Tibshirani, 2003*).

For comparison to epiblast and extraembryonic ectoderm, WGBS data was used and only CpGs that overlapped with RRBS-captured CpGs were analyzed.

## ChIP-seq and ChIP-BS data analysis

Raw sequencing reads were aligned to the mm9 build of the mouse genome using Bowtie 2 (*Langmead and Salzberg, 2012*). Browser tracks were created using igvtools and visualized using the Integrative Genomics Viewer (*Robinson et al., 2011*; *Thorvaldsdóttir et al., 2013*). Peak calling was done using MACS2 with standard settings (*Zhang et al., 2008*). Promoters and CGIs were classified as being marked by H3K4me3 or H3K27me3 if they overlapped with the peaks called from the corresponding ChIP-seq dataset. Reads per kilobase per million mapped reads (RPKM) values were calculated using a custom Python script.

The number of reads for CGIs were calculated using a 1 kb window for H3K4me3 and a 5 kb windows for H3K27me3. The global distribution of read counts was fitted to a Poisson distribution to calculate *P*-values, and *q*-values are calculated using the R qvalue package (*Storey and Tibshirani, 2003*).

For ChIP-BS-seq, we used the same Python processing pipeline described above for RRBS.

## RNA-seq data analysis

Raw sequencing reads were aligned to the mm9 build of the mouse genome and with Ensembl gene models (Ensembl release 67) using Tophat 2 (*Kim et al., 2013*). Reads overlapping gene models were counted using featureCounts (*Liao et al., 2014*). Differential expression analysis was done with DESeq2 (*Love et al., 2014*).

## DMR prediction

Liver DNase (ENCODE), H3K4me3, H3K27me3, H3K36me3 (ENCODE) enrichment for each CGI were calculated using the log(RPKM +0.001), normalized to *z*-scores. RPKM values higher than the 99[th] percentile were filtered out. Only CGIs with starting methylation less than 0.2 were retained in the analysis, to create a set of canonical, unmethylated CGIs. DMR prediction was done using the

following R packages: glm for logistic regression, e1071 for naïve Bayes, rpart for decision tree, randomForest for random forest, xgboost for gradient-boosted trees. For random forest, the following parameters were used: 'classwt = c(1, nneg/npos), ntree = 100', where nneg and npos were the number of negative and positive examples respectively. For xgboost, the following parameters were used: 'objective = 'binary:logistic', max_depth = 10, nrounds = 10, scale_pos_weight = nneg/npos, subset = 0.7'. All other packages used the default parameter settings.

## Motif analysis

Find Individual Motif Occurrences (FIMO, (*Grant et al., 2011*)) was used to identify transcription factor motif presence within the Uncx amplicon sequence. The software outputs q-values that predict the likelihood of transcription factor binding based on its predicted binding motif sequence. This list of transcription factors was then filtered for only those expressed in MEFs (FPKM $\geq$10, $n$ = 11).

## Acknowledgments

Thanks to all members in Meissner lab for the useful discussion. AM is a New York Stem Cell Foundation, Robertson Investigator. This work was supported by NIH grant R01DA036898, the New York Stem Cell Foundation and the Max Planck Society.

## Additional information

### Funding

| Funder | Grant reference number | Author |
| --- | --- | --- |
| National Institutes of Health | R01DA036898 | Alexander Meissner |
| Max-Planck-Gesellschaft | | Alexander Meissner |
| New York Stem Cell Foundation | | Alexander Meissner |

The funders had no role in study design, data collection and interpretation, or the decision to submit the work for publication.

### Author contributions

Yingying Zhang, Conceptualization, Investigation, Writing—original draft, Writing—review and editing; Jocelyn Charlton, Validation, Investigation, Methodology, Writing—review and editing; Rahul Karnik, Data curation, Software, Formal analysis, Validation, Investigation, Visualization, Writing—review and editing, Writing—original draft; Isabel Beerman, Data curation, Investigation; Zachary D Smith, Resources, Investigation, Methodology, Writing—original draft, Writing—review and editing; Hongcang Gu, Methodology, Validation; Patrick Boyle, Xiaoli Mi, Kendell Clement, Andreas Gnirke, Resources, Validation, Methodology; Ramona Pop, Resources; Derrick J Rossi, Supervision, Validation, Investigation; Alexander Meissner, Conceptualization, Supervision, Funding acquisition, Validation, Project administration, Writing—original draft, Writing—review and editing

### Author ORCIDs

Rahul Karnik https://orcid.org/0000-0002-0400-635X
Kendell Clement http://orcid.org/0000-0003-3808-0811
Alexander Meissner https://orcid.org/0000-0001-8646-7469

### Ethics

Animal experimentation: This study was performed in accordance with the recommendations in the Guide for the Care and Use of Laboratory Animals of the National Institutes of Health. All of the animals were handled according to approved institutional animal care and use committee (IACUC) protocols (#28-21) of Harvard University.

Decision letter and Author response
Decision letter https://doi.org/10.7554/eLife.40757.023
Author response https://doi.org/10.7554/eLife.40757.024

# Additional files

## Supplementary files

• Supplementary file 1. Sequence of primers used for RT-qPCR and bisulfitesequencing.
DOI: https://doi.org/10.7554/eLife.40757.018

• Transparent reporting form
DOI: https://doi.org/10.7554/eLife.40757.019

## Data availability

Data have been uploaded to GEO under GSE117909.

The following dataset was generated:

| Author(s) | Year | Dataset title | Dataset URL | Database and Identifier |
|---|---|---|---|---|
| Alexander Meissner, Yingying Zhang, Jocelyn Charlton, Rahul Karnik, Zachary D Smith, Andreas Gnirke | 2018 | Targets and genomic constraints of ectopic Dnmt3b expression | https://www.ncbi.nlm.nih.gov/geo/query/acc.cgi?acc=GSE117909 | Gene Expression Omnibus, GSE117909 |

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
