## [Decision Letter]

[Editors’ note: a previous version of this study was rejected after peer review, but the authors submitted for reconsideration. The first decision letter after peer review is shown below.]

Thank you for submitting your work entitled "Targets and genomic constraints of ectopic de novo DNA methyltransferase expression" for consideration by *eLife*. Your article has been reviewed by three peer reviewers, and the evaluation has been overseen by a Reviewing Editor and a Senior Editor. The following individual involved in review of your submission have agreed to reveal their identity: Albert Jeltsch (Reviewer #2).

Our decision has been reached after consultation between the reviewers. Based on these discussions and the individual reviews below, we regret to inform you that your work will not be considered further for publication in *eLife*.

Studying the contribution of DNA methyltransferases in gene regulation is an interesting and relevant topic in the field of epigenetics. In this study authors have studied the consequences of DNMT3 over expression in a mouse model in an effort to understand what goes astray in cancer cells where DNMT3 is frequently misregulated. Authors show that upon DNMT3B overexpression certain genomic regions are more sensitive such as silent, H3K27me3-enriched CGI targets compared to bivalent regions or regions enriched for H3K4me3.

Overall the reviewers are mixed. Although the reviewers appreciated that experiments are thoroughly performed, concerns were raised as to the functional relevance of the findings to the cancer model. Moreover, the experiments presented lack sufficient mechanistic insights to unravel the functional relationship between DNA methylation and H3K27me3/polycomb mediating silencing. In the light of these concerns, we cannot accept the manuscript but would be willing to review an extensively revised version of the manuscript.

*Reviewer #1:*

In normal somatic tissues DNMT3A is more highly expressed than its counterpart, DNMT3B. However, in human cancers DNMT3B is often overexpressed-more so than DNMT3A. Concordantly, the methylomes of cancers are almost universally misregulated. In this study, Zhang and colleagues set out to establish an in vivo model to investigate the molecular events that lead to aberrant DNA methylation patterns using a mouse model in which an inducible promoter drives Dnmt3B. This permits for the flexibility to analyze Dnmt3B overexpression in a tissue-by-tissue manner as well as in a developmental context, to investigate the dynamics and cellular heterogeneity of aberrant DNA methylation gain and to address its causative role in abnormal cell homeostasis/tumorigenesis.

The major takeaway from the study is that when Dnmt3B is overexpressed, DNA methylation tend to target silent, H3K27me3-enriched CGI targets. Then, more specifically, the study shows that i) some CpG sites are more prone to immediate de novo DNA methylation than others; ii) ectopic DNMT3B promotes de novo DNA methylation by transient associations; and iii) DNMT3B-induced DNA methylation can initially co-occur with H3K27me3, before the latter depletes. The main finding was rather intuitive based on (1) the known antagonistic interplay between active chromatin (namely H3K4me3) and DNMT3 enzymes, and (2) a litany of studies (many of which were cited here) that implicates polycomb-silenced loci as predisposed for aberrant hyper DNA methylation in cancers. Given the spin of this paper was its implication for cancer models, very little investigation was performed into how such ectopic DNA methylation may promote cancer-like properties, nor is a mechanistic underpinning provided for the polycomb-DNA methylation relationship: it is still unknown why bivalent DNA methylation/H3K27me3 states are unstable (or how DNA methylation deposition prevents H3K27me3 maintenance) and why not all H3K27me3-occupied CGIs gain DNA methylation upon Dnmt3B overexpression. While the science performed to carry out this study was largely sound, this looks like a preliminary (and slightly superficial) characterization of a Dnmt3B-overexpressing mouse.

Below are detailed my major comments:

1) Although the authors give the impression that they will be able to induce Dnmt3B overexpression tissue-by-tissue, they have only performed a systemic induction, in the entire animal, through doxycycline supplied in drinking water for 3 months. This should be more explicit in the first paragraph of the Results section.

2) The Eed KD section is too brief and glossed over. To wit, there must be a subset of genes that become ectopically expressed in the absence of PRC2. These would then be marked by K4me3, and impervious to DNA methylation. Therefore, I am surprised that the DMRs are so concordant ("virtually identical" subsection “Loss of H3K27me3 does not change the target spectrum of Dnmt3b”) between Control and Eed shRNA samples. This section may benefit from an accompanying RNA-seq analysis from these same Eed KD to assess which genes are activated. If none, then it may mean that the proper transcription factors are not present in MEFs, and this should be commented.

3) Some 70% (and therefore the majority) of H3K27me3-marked CGIs are not targets for aberrant methylation (Figure 3G), so what confers the protection? Is it correlated with the density of the polycomb-domains? Co-existence of the chromatin marks? If most of these regions remained unexpressed without PRC2, I would anticipate they would be readily accessed by DNMT3B. If this large fraction remains unmethylated, that is interesting in and of itself, as there must be some other component of chromatin that is refractory to de novo DNA methylation (KDM2B/FBXL10?). This deserves further investigation or at least, commentary.

4) It seems important to verify, at least for a few loci, that Dnmt3b-induced gain of DNA methylation quantified by RRBS corresponds to methylated but not hydroxymethylated cytosines.

5) In Figure 5, the authors found, at a single locus, that some CpG sites may be more sensitive to DNA methylation gain than others. No further insight into the origin of this differential treatment of some CpG sites versus others (sequence context? Motifs for transcription factors?). I do not see the point of mentioning it without a more thorough (and likely genome-wide) analysis of this specificity of action.

*Reviewer #2:*

This is a very interesting paper addressing one of the most burning questions in Molecular Biology, namely the dynamics of the establishment of complex epigenetic patterns. The authors collected high-end experimental data including several controls. In general, the experiments were conducted very carefully, data are presented clearly, and the interpretation is convincing. In my view this paper is ideally suited for *eLife*, probably being at the upper quality end of papers published in this journal. I have collected a couple of questions and comments that may help the authors to further improve this very nice manuscript.

1) Data must be made available and this must be clarified in the manuscript.

2) General comment: For non-specialist it may be helpful to mention that the H3K4me3 effect is mediated by the ADD domain of Dnmt3, hence it is a direct effect. It may also be helpful to mention that the enrichment in gene bodies is mediated by H3K36me3 binding to PWWP.

3) I found several places in which results were described in present tense. While I did not go into the detailed grammatical structure of each sentence, I found many places where past tense should have been used. The authors should critically check this and correct as needed.

4) We have 3 styles of data presentation, Figure 1C, Figure 2B and Figure 3B. I recommend to change Figure 3B to Figure 2B style. Also, in the transition of the experiments shown in Figure 1 to those in Figure 2, it would be nice to show the Figure 2B data in a Figure 1C like style, to allow for a comparison.

5) In Figure 2E, what is the p-value for the discrimination of the Fcgr3 CLP and CD4^+^ results. Does this justify the blue box?

6) Figure 1—figure supplement 4 and its legend are unclear. What is shown DNMT3A and 3B overexpression? But why is there one control? Please check.

7) Figure 4E shows enrichment of K27me3 at CGIs. How does this look, if DMRs are used?

8) I did not understand Figure 4G. Also, I do not see that what is described in the legend is shown, sorry.

9) A genome-wide analysis of the data from Chip-Bislfite should be presented.

10) Subsection “Dnmt3b expression triggers extensive heterogeneity in DNA methylation patterns at previously unmethylated loci”: It may be useful to mention that Dnmt3 enzymes are known to have very strong flanking sequence preferences. The discussion goes in this direction, but this may be made more explicit.

11) The authors argue there is little correlation in Figure 5D. I am not fully convinced by this statement, because Figure 5B seems to show a different thing. Did the authors scale correlation by methylation levels? This is relevant, because co-occurrence of two rare events is much more meaningful. I think in this part a more sophisticated analytical approach is needed to support the claims.

12) The data shown in Figure 5E and f are the only really weak point of the manuscript. The experiment is unclear, because the mutant contains more than one alteration, the level of controls is lower and after all, the data do not contribute much. In my view the paper would be stronger after omission of these data.

*Reviewer #3:*

In this manuscript, Zhang et al., revealed that H3K27me3-enrich CGIs are prone to hyper-DNA methylation when DNMT3B is ectopically overexpressed. Using DNMT3b-inducible mice, the authors showed a large number of CGIs exhibited hyper DNA-methylation in the three tissues (intestine, liver, and kidney) of the mice. Moreover, the analysis of correlation between the DMR and their histone modification status showed H3K27me3-only CGIs were susceptible to gain DNA methylation than those that are bivalent or enrich for H3K4me3 in liver and fibroblast isolated from the transgenic mice. Nonetheless, Eed knockdown that can reduce global H3K27me3 levels did not affect target spectrum of DNMT3B, suggesting H3K27me3 itself was not a driving factor for hypermethylation induced by ectopic DNMT3B expression. Lastly, based on the results indicating DNMT3B was much less enriched in hypermethylated regions, they proposed that DNMT3B methylated target loci by transient associations with DNA in a hit-and-run fashion.

Issues that should be clarified:

1) Figure 1; to what extent is the hypermethylation induced by DNMT3B overexpression similar to aberrant methylation observed in some of cancers? Does annotation of aberrant methylation loci in cancer show similar results to that of DNMT3B OE as shown in Figure 1—figure supplement 6? This is likely important for distinguish whether ectopic DNMT3B expression system is useful to study the link between cancer and aberrant DNA methylation.

2) Figure 1—figure supplement 3 and Figure 1—figure supplement 4; there is no description about DNMT3A-inducible mouse.

3) Figure 1—figure supplement 4; there is no data of DNMT3A expression levels, even though it's mentioned in figure legend.

4) Figure 3; as shown in previous reports (Oncotarget (2017); Genome Research (2012)), bivalent CGIs are known to be main target of aberrant hypermethylation in cancer. In contrast, H3K27me3-only CGIs were susceptible to gain DNA methylation by DNMT3B OE. As mention above, to what extent are hypermethylation loci by DNMT3B OE similar to aberrant methylation loci in cancers from the viewpoint of correlation between histone modification and acquired DNA methylation? The authors should discuss this in more detail.

[Editors’ note: what now follows is the decision letter after the authors submitted for further consideration.]

Thank you for submitting your article "Targets and genomic constraints of ectopic Dnmt3b expression" for consideration by *eLife*. Your article has been reviewed by three peer reviewers, and the evaluation has been overseen by a Reviewing Editor and Jessica Tyler as the Senior Editor. The following individuals involved in review of your submission have agreed to reveal their identity: Albert Jeltsch (Reviewer #2).

The reviewers have discussed the reviews with one another and the Reviewing Editor has drafted this decision to help you prepare a revised submission.

The reviewers all agree that you have responded to their previous concerns comprehensively and convincingly. However, there remain a small number of remaining comments raised by the reviewers in their reviews (copied below.

*Reviewer #1:*

In the first round of review, my main concerns were related to a lack of depth in the analyses and blurry connection with cancer, where DNA methylation is also often gained at the expense of H3K27me3 consequently to DNMT3B overexpression.

Although the authors did not provide any of the additional experiment we suggested, they have now included more thorough analyses and comparison with relevant in vivo systems, namely cancer cells and extraembryonic ectoderm. To me, the most provocative message is that DNA methylation and H3K27me3 can transiently co-occur at CGIs before H3K27me3 depletes, and that this may not be linked to a repulsive effect of the DNA methylation mark per se but through the physical interference that DNMT3B may exert on PRC2 recruitment. Does it mean that H3K27me3 may similarly deplete at sensitive CGIs upon overexpression of a catalytically dead DNMT3B?

Anyway, this study is well conceived and involves an impressive set of analyses. It may bring more questions than answers, but this will certainly stimulate novel investigation paths in the field.

*Reviewer #2:*

This is a very interesting and technically sound paper. The authors have convincingly responded to my previous comments. However, I have collected some concerns related to the new version that deserve the attention of the authors.

Major points:

1) There are two clear chromatin parameters with obvious connection to DNMT3B targeting which have not been considered and I wonder why this is the case. It is clear that H3K36 methylation recruits DNMT3 enzymes via their PWWP domain. Moreover, in the relatively artificial setting studied here the accessibility of chromatin as seen by ATAC-seq or any related approach may help to discriminate good and bad targets. Data sets should be available, and I suggest to include these parameters into the analysis at least for some example tissues or cell lines.

2) Subsection “Knockdown of Eed has a limited impact on MEF CGI hypermethylation”: The EED KD does not reduce H3K27me3 to zero. Moreover, cells likely will contain the same pattern of H3K27me3, just at lower overall levels. Based on this, the statement that H3K27me3 and PRC2 are not needed for induction of hypermethylation cannot be made, not even in the toned down form as in the paper. This question is simply unanswered.

3) The term "hit-and-run" in the abstract is unclear. Do the authors imply to state "distributively", which would be the correct term in molecular enzymology? Based on my next arguments this sentence likely should be removed from the Abstract.

4) Figure 4D and the corresponding subsection” Dnmt3b expression induces local methylation discordance”. The primary observation is fine and well documented- there is no correlation of methylation states of adjacent sites. Unfortunately, things are more complicated than anticipated by the authors. One may imagine a model of processive DNA methylation in which the DNMT binds DNA, moves along the DNA and methylates sites once a while here and there, but not each and every site on its track. If many sites are left unmodified during this random walk along the DNA, the final pattern would look like the one observed by the authors. Still it would arise from a fully processive enzyme that perhaps never dissociates from the DNA. Hence the last sentence of subsection “Dnmt3b expression induces local methylation discordance” should be removed. As stated above, the term "hit-and-run" is anyway not appropriate. Based on the more limited conclusions that can be drawn from this analysis, it might be moved to the supplement entirely.

5) I understand that the paragraph in subsection “Ectopic Dnmt3b targets shares features with cancer CGI hypermethylation” was partially stimulated by reviewer #1, but I am not very convinced by this. First of all, tumors are the outcome of heavy selection, which has been shown to have a pronounced effect on methylation profiles (PMID: 28215704). Secondly, Figure 6E clearly shows that DNMT3B OE and cancer methylomes differ substantially. The conclusions in this part should be toned down heavily. I see the main value and important contribution of this work in basic research unravelling the pathways and mechanisms of how DNMT3B approaches DNA and how the chromatin modification network is organized.

*Reviewer #3:*

The authors have made a number of revisions to the manuscript that, in my opinion, comprehensively address the issues raised by three reviewers. The important revisions are shown below:

In new Figure 6, to assess whether the CGI hypermethylation observed in 3b OE tissues is similar to aberrant methylation in cancers, they identified and characterized CGIs in which DNA methylation is induced by Dnmt3b OE and compared these to abnormally methylated CGIs observed in human CLL. They found that most H3K27me3-only and bivalent CGIs are ectopically methylated in Dnmt3b OE mouse model, while a much smaller proportion are methylated in human CLL. Further, global hypomethylation was not observed in Dnmt3b OE system. Thus, they concluded that although their ectopic system shares some targets and possible mechanisms with human cancer, it does not fully recapitulate the local and global methylation trends that characterize the cancer methylome. In addition, based on these results, the authors revised main text in results and discussion to tone down their arguments on the utility of Dnmt3b OE system and its relationship to cancer. In my opinion, it made the manuscript more accurate and evidence-based.

---

## [Author Response]

[Editors’ note: the author responses to the first round of peer review follow.]

[…] Reviewer #1:*In normal somatic tissues DNMT3A is more highly expressed than its counterpart, DNMT3B. However, in human cancers DNMT3B is often overexpressed-more so than DNMT3A. Concordantly, the methylomes of cancers are almost universally misregulated. In this study, Zhang and colleagues set out to establish an* in vivo model to investigate the molecular events that lead to aberrant DNA methylation patterns using a mouse model in which an inducible promoter drives Dnmt3B. This permits for the flexibility to analyze Dnmt3B overexpression in a tissue-by-tissue manner as well as in a developmental context, to investigate the dynamics and cellular heterogeneity of aberrant DNA methylation gain and to address its causative role in abnormal cell homeostasis/tumorigenesis.

*The major takeaway from the study is that when Dnmt3B is overexpressed, DNA methylation tend to target silent, H3K27me3-enriched CGI targets. Then, more specifically, the study shows that i) some CpG sites are more prone to immediate* de novo *DNA methylation than others; ii) ectopic DNMT3B promotes* de novo *DNA methylation by transient associations; and iii) DNMT3B-induced DNA methylation can initially co-occur with H3K27me3, before the latter depletes. The main finding was rather intuitive based on (1) the known antagonistic interplay between active chromatin (namely H3K4me3) and DNMT3 enzymes, and (2) a litany of studies (many of which were cited here) that implicates polycomb-silenced loci as predisposed for aberrant hyper DNA methylation in cancers. Given the spin of this paper was its implication for cancer models, very little investigation was performed into how such ectopic DNA methylation may promote cancer-like properties, nor is a mechanistic underpinning provided for the polycomb-DNA methylation relationship: it is still unknown why bivalent DNA methylation/H3K27me3 states are unstable (or how DNA methylation deposition prevents H3K27me3 maintenance) and why not all H3K27me3-occupied CGIs gain DNA methylation upon Dnmt3B overexpression. While the science performed to carry out this study was largely sound, this looks like a preliminary (and slightly superficial) characterization of a Dnmt3B-overexpressing mouse.*

We thank the reviewer for the very careful reading of our manuscript and the many detailed comments, which we have addressed in the point-by-point responses below. As outlined below we have tried to make the value of the system in connection to cancer biology more clear (updated Figure 1 and new Figure 6) while at the same time trying to balance its utility and not oversell it (see text in the Results section and Discussion section). Other analyses were expanded, and most figures improved in their display to address the preliminary nature raised.

Essential revisions:1) Although the authors give the impression that they will be able to induce Dnmt3B overexpression tissue-by-tissue, they have only performed a systemic induction, in the entire animal, through doxycycline supplied in drinking water for 3 months. This should be more explicit in the first paragraph of the Results section.

We apologize if it was unclear that the Dnmt3b overexpression is systemic in the transgenic mice. As the reviewer suggested, we have updated our Results section to explicitly state that the induction of Dnmt3b was through drinking water supplied doxycycline.

2) The Eed KD section is too brief and glossed over. To wit, there must be a subset of genes that become ectopically expressed in the absence of PRC2. These would then be marked by K4me3, and impervious to DNA methylation. Therefore, I am surprised that the DMRs are so concordant ("virtually identical" subsection “Loss of H3K27me3 does not change the target spectrum of Dnmt3b”) between Control and Eed shRNA samples. This section may benefit from an accompanying RNA-seq analysis from these same Eed KD to assess which genes are activated. If none, then it may mean that the proper transcription factors are not present in MEFs, and this should be commented.

The reviewer is pointing out that at bivalent promoters, loss of H3K27me3 would leave H3K4me3 alone, which could alter gene expression. In fact, previous work shows that loss of PRC2 recruitment does not have a substantial affect on gene expression (Riising et al., 2014), likely due to lack of expression of the transcription factors required to activate these genes (as the reviewer suggests). However, it is an interesting point that differential targeting may be observed at bivalent domains, and we therefore went back to our Eed KD RRBS data to explore this in further detail. As described, we used two different shRNAs to target Eed, and upon dox treatment, the first showed overall similar CGI targeting and mean methylation to WT Eed MEFs (no KD: *n* = 5,943, mean = 0.22, shRNA4: *n* = 5,883, mean = 0.21). Interestingly, the second shRNA did show reduced methylation of CGIs (*n* = 4,510, mean = 0.16) along with increased Eed KD (qPCR in Figure 4—figure supplement 2C) and reduced global H3K27me3 (Western blot in Figure 4—figure supplement 2B). This points to a possible dependency that would be similar to what we recently observed in the early embryo (Smith et al., 2017). We now display this data as a heatmap instead of a PCA plot to simplify understanding (new Figure 4G). We next identified CGIs that were differentially methylated between our control (no dox) samples and found a small number that were preferentially located in regions with no H3K4me3 or H3K27me3. In contrast, when comparing CGI methylation between our dox treated samples, we found that CGIs methylated in the WT Eed MEFs, that were not targeted once Eed was knocked down, were overlapping with bivalent chromatin (new Figure 4G-H). We have therefore expanded our Discussion section on this.

*3) Some 70% (and therefore the majority) of H3K27me3-marked CGIs are not targets for aberrant methylation (Figure 3G), so what confers the protection? Is it correlated with the density of the polycomb-domains? Co-existence of the chromatin marks? If most of these regions remained unexpressed without PRC2, I would anticipate they would be readily accessed by DNMT3B. If this large fraction remains unmethylated, that is interesting in and of itself, as there must be some other component of chromatin that is refractory to* de novo *DNA methylation (KDM2B/FBXL10?). This deserves further investigation or at least, commentary.*

We fully agree with the reviewer that this is indeed an interesting point, and we currently do not have an explanation for why only a subset of H3K27me3-marked CGIs would be targeted by ectopic Dnmt3b expression. This is also the case for the extraembryonic ectoderm, which shows CGI hypermethylation that only affects a subset of PRC2 targets. And this also holds true for human cancers where most CGIs remain unmethylated despite being PRC2 targets. Therefore, PRC2 clearly does not define targeting alone, and there are yet to be discovered factors that guide the de novo Dnmts to selected targets. We have added more to our discussion to expand on this point and suggest the need for further follow up studies.

4) It seems important to verify, at least for a few loci, that Dnmt3b-induced gain of DNA methylation quantified by RRBS corresponds to methylated but not hydroxymethylated cytosines.

To explore this, we randomly selected 6 CpG sites that display low or high methylation levels in bulk RRBS from the liver of adult mice after dox induction for 3 months and checked the hydroxymethylation level of them using the Quest 5-hmC Detection Kit (Zymo Research). No significant increase in hydroxymethylation level was observed for these CpG sites in the liver with Dnmt3b overexpression compared to the control mice. We have added this data as new Figure 1—figure supplement 3D – and updated the manuscript accordingly on page 4. Although limited to a few sites we feel that this is sufficient together with our revised text to describe what we observe.

5) In Figure 5, the authors found, at a single locus, that some CpG sites may be more sensitive to DNA methylation gain than others. No further insight into the origin of this differential treatment of some CpG sites versus others (sequence context? Motifs for transcription factors?). I do not see the point of mentioning it without a more thorough (and likely genome-wide) analysis of this specificity of action.

We agree that this deserved more analysis. First, we have added new Figure 5—figure supplement 1B that shows the presence of various transcription factor motifs at the selected locus, with the added condition that these transcription factors are expressed in MEFs. Since this analysis suggested that the presence of an Sp1 motif might protect against the gain of methylation, we looked at Sp1 sites at CGIs that gain methylation overall and found that the trend was observed genome-wide as well (new Figure 5—figure supplement 1C). We have updated the manuscript accordingly.

Reviewer #2:[…] 1) Data must be made available and this must be clarified in the manuscript.

We fully agree and have deposited our data in GEO under accession number: GSE117909 We have also added the accession code to the manuscript and include a private reviewer link so that the reviewer can already access the data: https://www.ncbi.nlm.nih.gov/geo/query/acc.cgi?acc=GSE117909

Use token: ihedwsgcbbkrvud

2) General comment: For non-specialist it may be helpful to mention that the H3K4me3 effect is mediated by the ADD domain of Dnmt3, hence it is a direct effect. It may also be helpful to mention that the enrichment in gene bodies is mediated by H3K36me3 binding to PWWP.

We thank the reviewer for these suggestions and we have expanded the text accordingly and also added the following references (Ooi et al., 2007; Otani et al., 2009; Zhang et al., 2010).

3) I found several places in which results were described in present tense. While I did not go into the detailed grammatical structure of each sentence, I found many places where past tense should have been used. The authors should critically check this and correct as needed.

Thank you for this feedback, we have carefully checked the manuscript for consistent tense and style.

4) We have 3 styles of data presentation, Figure 1C, Figure 2B and Figure 3B. I recommend to change Figure 3B to Figure 2B style. Also, in the transition of the experiments shown in Figure 1 to those in Figure 2, it would be nice to show the Figure 2B data in a Figure 1C like style, to allow for a comparison.

While we agree with the reviewer’s suggestion to show the data in Figure 2B as in Figure 1C, and have added this as new Figure 2B for easy comparison to the data displayed in Figure 1, we feel that changing Figure 3B to the style of (old) Figure 2B may be misleading, as in (old) Figure 2B we display CGI methylation in 3B OE compared to control, in 3B we instead incorporate change in expression levels (FPKM) and wouldn’t want a reader to mistake this for methylation.

5) In Figure 2E, what is the p-value for the discrimination of the Fcgr3 CLP and CD4^+^ results. Does this justify the blue box?

For the CGIs located near to *Fcgr3,* comparing control and 3b OE CLP samples generated highly significant *P* values (2.5 x 10^-11^ to 4.03 x 10^-22^). However, although the level of methylation gained at CGIs in CD4^+^ cells was clearly lower than in CLP cells (mean difference of 0.19 vs 0.37), the *P* values for CD4^+^ 3bOE vs control were still significant (6.0 x 10^-4^ to 7.83 x 10^-6^). In this figure, we use the blue box to display the difference in overall methylation gain and have clarified both the level of significance as well as the color schematic in the figure legend.

6) Figure 1—figure supplement 4 and its legend are unclear. What is shown DNMT3A and 3B overexpression? But why is there one control? Please check.

We apologize that the figure was unclear. Figure 1—figure supplement 3 (now Figure 1—figure supplement 3C) has now been edited so that it shows the expression of Dnmt3b in intestine, liver and kidney from control and 3b OE mice. We have also updated the figure legend.

7) Figure 4E shows enrichment of K27me3 at CGIs. How does this look, if DMRs are used?

Figure 4E shows enrichment of K27me3 ChIP-BS at CGIs, and this figure already uses DMRs (combined for day 1 and day 7) – we have added this to the figure to make this clear and edited

8) I did not understand Figure 4G. Also, I do not see that what is described in the legend is shown, sorry.

We apologize and agree that this figure was difficult to interpret. We have now re-analyzed this data (as described in response to reviewer 1 point 2) and have updated new Figure 4G-H. We hope this is now easier to understand.

9) A genome-wide analysis of the data from Chip-Bislfite should be presented.

We apologize for not making this more clearly – Figures 4D and 4E show the genome-wide analysis of the ChIP-bisulfite sequencing data. We have edited the figure legends to clarify the ChIP-BS-seq data was used for these panels.

10) Subsection “Dnmt3b expression triggers extensive heterogeneity in DNA methylation patterns at previously unmethylated loci”: It may be useful to mention that Dnmt3 enzymes are known to have very strong flanking sequence preferences. The discussion goes in this direction, but this may be made more explicit.

We agree that it is interesting to add this and have included the relevant discussion and references in the main text in subsection “Ectopic Dnmt3b targets shares features with cancer CGI hypermethylation”.

11) The authors argue there is little correlation in Figure 5D. I am not fully convinced by this statement, because Figure 5B seems to show a different thing. Did the authors scale correlation by methylation levels? This is relevant, because co-occurrence of two rare events is much more meaningful. I think in this part a more sophisticated analytical approach is needed to support the claims.

We understand that Figure 5B shows that some neighboring CpGs have similar susceptibility to gain methylation, and this phenomenon can perhaps be explained by the presence of transcription factor motifs as shown in Figure 5—figure supplement 1B. However, Figure 5D calculates correlations using the per-read methylation level at each CpG, which is either 0 or 1 only. So, while some neighboring CpGs have similar average methylation levels across the cell population as seen in Figure 5B, Figure 5D shows that at the individual read level, neighboring CpGs show little correlation, which suggests that the DNA methylation deposited by DNMT3B is not processive. To make this clearer we have extended our discussion on this in the Discussion section.

12) The data shown in Figure 5E and f are the only really weak point of the manuscript. The experiment is unclear, because the mutant contains more than one alteration, the level of controls is lower and after all, the data do not contribute much. In my view the paper would be stronger after omission of these data.

We thank the reviewer for this suggestion. We have removed this part from the current version of the manuscript.

Reviewer #3:[…] 1) Figure 1; to what extent is the hypermethylation induced by DNMT3B overexpression similar to aberrant methylation observed in some of cancers? Does annotation of aberrant methylation loci in cancer show similar results to that of DNMT3B OE as shown in Figure 1—figure supplement 6? This is likely important for distinguish whether ectopic DNMT3B expression system is useful to study the link between cancer and aberrant DNA methylation.

The reviewer is right on target here and this was part of our original motivation to explore whether this ectopic model could provide mechanistic insights into the cancer methylome. Our results make several relevant points to interpret and utilize the model appropriately. We first need to point out that from histological examination of our tissues, we did not see any evidence of tumour formation (Figure 1—figure supplement 4A). However, this was in a relatively short time frame (3 months of dox) and these mice may have developed cancer if left for a longer time period, as previously observed (Linhart et al., 2007).

To assess whether the CGI hypermethylation observed in 3b OE tissues is similar to aberrant methylation in cancers, we took all orthologous CGIs that could be mapped in both human and mouse and studied methylation levels in mouse kidney, liver and B-cells as well as human haematopoetic cancer CLL. We also included the mouse extraembryonic ectoderm (ExE) and age-matched epiblast in this comparison, as we have recently shown that CGIs hypermethylated in the ExE are also methylated across a wide range of cancers (Smith et al., 2017). This interesting finding suggests that the cancer methylome mirrors features of an early developmental stage where characteristics observed in cancer are also required (angiogenesis, proliferation, invasion). This analysis showed high concordance between CGIs methylated across all tissues (new Figure 6—figure supplement 1B). To more closely examine this, we focused on mouse normal and 3b OE B cells and compared these to human normal B cells and CLL. We separated CGIs based on their chromatin modifications in normal B cells and compared the overlap between targeted CGIs. This analysis showed that most H3K27me3-only and bivalent CGIs are ectopically methylated in our mouse model, while a much smaller proportion are methylated in human CLL. Furthermore, we have recently shown that CGI gain in cancer is associated with an increase in methylation discordance (or percentage of discordant reads) (Landau et al., 2014). We show this similarity in our manuscript as an increase of epialleles after doxycycline treatment in MEFs (current Figure 5C). We have now added a new analysis to compare the methylation discordance between control and induced samples as well as between human B cell and CLL. In all three cases, an increase in the number of reads with discordant neighboring methylation states was observed, indicating a comparable gain in discordance (new Figure 6—figure supplement 2A). Although we see similar features of CGI targets, cancers (and the ExE) are characterized by global hypomethylation (new Figure 6E and Figure 6—figure supplement 2B). In our system, we do not see this global hypomethylation (new Figure 6—figure supplement 2B) and hence have a useful system where the two (usually connected) features of cancer types can be investigated in isolation. In summary, we believe that our ectopic Dnmt3b OE system is a suitable model for studying some aspects of the hypermethylation at CGIs, but we clearly observe a higher frequency of targets, which could be associated with expression levels, or result from different targeting mechanisms.

2) Figure 1—figure supplement 3 and Figure 1—figure supplement 4; there is no description about DNMT3A-inducible mouse.3) Figure 1—figure supplement 4; there is no data of DNMT3A expression levels, even though it's mentioned in figure legend.

Both points 2 and 3 refer to the same figure: we apologize for these mistakes and have now removed “Dnmt3a” from the title of Figure 1—figure supplement 3. We have also remade Figure 1—figure supplement 4 (Figure 1—figure supplement 3C) to show only data for Dnmt3b OE mice.

4) Figure 3; as shown in previous reports (Oncotarget (2017); Genome Research (2012)), bivalent CGIs are known to be main target of aberrant hypermethylation in cancer. In contrast, H3K27me3-only CGIs were susceptible to gain DNA methylation by DNMT3B OE. As mention above, to what extent are hypermethylation loci by DNMT3B OE similar to aberrant methylation loci in cancers from the viewpoint of correlation between histone modification and acquired DNA methylation? The authors should discuss this in more detail.

In our study, we observed gain of methylation at both H3K27me3 targets as well as tissue specific bivalent domains. In the first reference, Court and Arnaud, (2017) identify a list of bivalent promoters in human ESCs and find that these were preferential targets for CGI hypermethylation across 8 different tumor types. To test whether these might also be targets in our system, we identified the orthologous sites and compared methylation levels between these and all CGIs in control and 3b OE intestine, kidney and liver. Here, we saw in every tissue that the orthologous CGIs were more highly methylated than background (new Figure 6—figure supplement 1C). Upon further examination of all CGIs that can be mapped in both human and mouse genomes, we see that ~40% CGIs are located in bivalent chromatin in human ESCs. We then calculated the proportion of CGIs gaining hypermethylation that were located in hESC bivalent domains and found positive enrichment for all samples examined (kidney, liver, MEFs and B-cells as well as human haematopoetic cancer CLL – new Figure 6—figure supplement 1C). This suggests that CGIs susceptible to hypermethylation are somewhat conserved between species. We have added this to the main text and extended our discussion accordingly.

[Editors' note: the author responses to the re-review follow.]

The reviewers have discussed the reviews with one another and the Reviewing Editor has drafted this decision to help you prepare a revised submission.The reviewers all agree that you have responded to their previous concerns comprehensively and convincingly. However, there remain a small number of remaining comments raised by the reviewers in their reviews (copied below).

We would like to thank the editors and three reviewers for the thoughtful handling of our manuscript and very much appreciate the time commitment. Below we respond to the last suggestions and have implemented them as indicated in the revised manuscript, which we hope is now suitable for publication.

Reviewer #1:In the first round of review, my main concerns were related to a lack of depth in the analyses and blurry connection with cancer, where DNA methylation is also often gained at the expense of H3K27me3 consequently to DNMT3B overexpression.

*Although the authors did not provide any of the additional experiment we suggested, they have now included more thorough analyses and comparison with relevant* in vivo *systems, namely cancer cells and extraembryonic ectoderm. To me, the most provocative message is that DNA methylation and H3K27me3 can transiently co-occur at CGIs before H3K27me3 depletes, and that this may not be linked to a repulsive effect of the DNA methylation mark per se but through the physical interference that DNMT3B may exert on PRC2 recruitment. Does it mean that H3K27me3 may similarly deplete at sensitive CGIs upon overexpression of a catalytically dead DNMT3B?*

Anyway, this study is well conceived and involves an impressive set of analyses. It may bring more questions than answers, but this will certainly stimulate novel investigation paths in the field.

We thank the reviewer for the additional feedback. We agree that this is not the end of the story and hope that our findings provide insight for other future studies. The roles of PRC2 and H3K27me3 are indeed very interesting, as is the idea that catalytically inactive DNMT3B may also interfere with binding of the PRC2 complex. We note that the catalytically inactive DNMT3B3 isoform has been shown to be reduced upon 5-aza treatment suggesting it is indeed able to interact with DNA (PMID:14757847). This binding may be important as it allows cofactors to be recruited to target loci – which was previously shown to stimulate gene body methylation (PMID:27121154) and induce methylation of specific loci in a cancer cell line (PMID:14757847). Although out of the scope of this study, we agree that it would be very interesting to determine the effects of overexpressing the catalytically inactive DNMT3B and have therefore expanded our discussion to include this point.

Reviewer #2:[…] 1) There are two clear chromatin parameters with obvious connection to DNMT3B targeting which have not been considered and I wonder why this is the case. It is clear that H3K36 methylation recruits DNMT3 enzymes via their PWWP domain. Moreover, in the relatively artificial setting studied here the accessibility of chromatin as seen by ATAC-seq or any related approach may help to discriminate good and bad targets. Data sets should be available, and I suggest to include these parameters into the analysis at least for some example tissues or cell lines.

We thank the reviewer for this suggestion. For mouse liver and kidney, H3K36me3 and DNase data sets were available from ENCODE. Regions enriched for H3K36me3 are generally highly methylated in control tissues already (mean 0.83 and 0.85 for kidney and liver respectively). However, following induction, these regions did increase further (mean 0.89 and 0.90). We have added a figure displaying this increase as new Figure 1—figure supplement 2C and updated the text in subsection “Ectopic Dnmt3b expression leads to widespread CpG island hypermethylation”.

To assess whether DNA accessibility was associated with methylation gain, we performed a random forest classifier to assess whether DMRs could be predicted based on eight different input features (new Figure 3D, Figure 3—figure supplement 1A). We then removed each feature in turn to assess the impact on performance and noted that H3K4me3 enrichment, DNase enrichment and methylation level in control tissue showed the greatest reduction (new Figure 3D, Figure 3—figure supplement 1B). These three parameters are highly correlated (new Figure 3—figure supplement 1C), whereby CGIs with almost no DNA methylation in control tissue, enriched for H3K4me3 in open chromatin are the least likely to gain methylation. We have added these findings to our manuscript and updated the text in subsection “H3K4me3 shields CGIs from aberrant DNA methylation”.

2) Subsection “Knockdown of Eed has a limited impact on MEF CGI hypermethylation”: The EED KD does not reduce H3K27me3 to zero. Moreover, cells likely will contain the same pattern of H3K27me3, just at lower overall levels. Based on this, the statement that H3K27me3 and PRC2 are not needed for induction of hypermethylation cannot be made, not even in the toned down form as in the paper. This question is simply unanswered.

We agree that unfortunately we are not entirely depleting H3K27me3 in our system. However, lower overall levels suggest some degree of change for H3K27me3 either across the genome or between cells in the population. If H3K27me3 was strictly required for hypermethylation, those cells/CGIs that did deplete H3K27me3 would no longer gain methylation, and we would expect to observe a reciprocal overall decrease in the number of CGIs that gain methylation, or lower mean methylation levels at targeted CGIs. Our Western shows a ~10-fold decrease in H3K27me3 levels, and as we still see a high level of targeting following induction as well as similar mean methylation levels, we believe it is fair to report these findings and at least suggest that H3K27me3 appears not essential for the CGI hypermethylation in this context. Given the reviewer’s concern, we have now re-written this section to clearly state our observations without drawing any strong conclusions, to let the reader interpret these findings for themselves. We hope that the reviewer is happy with our edit.

3) The term "hit-and-run" in the abstract is unclear. Do the authors imply to state "distributively", which would be the correct term in molecular enzymology? Based on my next arguments this sentence likely should be removed from the Abstract.

We have edited the term “hit-and-run” to “distributively” as suggested.

4) Figure 4D and the corresponding subsection “Dnmt3b expression induces local methylation discordance”. The primary observation is fine and well documented- there is no correlation of methylation states of adjacent sites. Unfortunately, things are more complicated than anticipated by the authors. One may imagine a model of processive DNA methylation in which the DNMT binds DNA, moves along the DNA and methylates sites once a while here and there, but not each and every site on its track. If many sites are left unmodified during this random walk along the DNA, the final pattern would look like the one observed by the authors. Still it would arise from a fully processive enzyme that perhaps never dissociates from the DNA. Hence the last sentence of subsection “Dnmt3b expression induces local methylation discordance” should be removed. As stated above, the term "hit-and-run" is anyway not appropriate. Based on the more limited conclusions that can be drawn from this analysis, it might be moved to the supplement entirely.

We believe that the reviewer is referring to Figure 5D which displays the absence of correlation between neighboring phased CpGs. We thank the reviewer for this feedback and have updated the text to include this possibility. We have also moved Figure 5D to the supplement as suggested.

5) I understand that the paragraph in subsection “Ectopic Dnmt3b targets shares features with cancer CGI hypermethylation” was partially stimulated by reviewer #1, but I am not very convinced by this. First of all, tumors are the outcome of heavy selection, which has been shown to have a pronounced effect on methylation profiles (PMID: 28215704). Secondly, Figure 6E clearly shows that DNMT3B OE and cancer methylomes differ substantially. The conclusions in this part should be toned down heavily. I see the main value and important contribution of this work in basic research unravelling the pathways and mechanisms of how DNMT3B approaches DNA and how the chromatin modification network is organized.

We appreciate this point and do not want to claim that our ectopic Dnmt3b OE system mimics cancer and in fact believe the difference is important to point out and discuss. We have therefore further edited the text to improve the conclusions and feel that it is valuable to describe all the possible targets for ectopic DNMT3 while noting that in cancer often only a subset is found. This conserved core set potentially suggests a common targeted activity that may be masked over time by additional targets arising through proliferation and selection.

Reviewer #3:The authors have made a number of revisions to the manuscript that, in my opinion, comprehensively address the issues raised by three reviewers. The important revisions are shown below:In new Figure 6, to assess whether the CGI hypermethylation observed in 3b OE tissues is similar to aberrant methylation in cancers, they identified and characterized CGIs in which DNA methylation is induced by Dnmt3b OE and compared these to abnormally methylated CGIs observed in human CLL. They found that most H3K27me3-only and bivalent CGIs are ectopically methylated in Dnmt3b OE mouse model, while a much smaller proportion are methylated in human CLL. Further, global hypomethylation was not observed in Dnmt3b OE system. Thus, they concluded that although their ectopic system shares some targets and possible mechanisms with human cancer, it does not fully recapitulate the local and global methylation trends that characterize the cancer methylome. In addition, based on these results, the authors revised main text in results and discussion to tone down their arguments on the utility of Dnmt3b OE system and its relationship to cancer. In my opinion, it made the manuscript more accurate and evidence-based.

We thank the reviewer for the positive feedback in response to our revision.